# A Near-optimal Algorithm for Learning Margin Halfspaces with Massart Noise

**Ilias Diakonikolas**
Department of Computer Sciences
UW-Madison
Madison, WI
ilias@cs.wisc.edu

**Nikos Zarifis**
Department of Computer Sciences
UW-Madison
Madison, WI
zarifis@wisc.edu

## Abstract

We study the problem of PAC learning $\gamma$-margin halfspaces in the presence of Massart noise. Without computational considerations, the sample complexity of this learning problem is known to be $\widetilde{\Theta}(1/(\gamma^2\epsilon))$. Prior computationally efficient algorithms for the problem incur sample complexity $\tilde{O}(1/(\gamma^4\epsilon^3))$ and achieve 0-1 error of $\eta + \epsilon$, where $\eta < 1/2$ is the upper bound on the noise rate. Recent work gave evidence of an information-computation tradeoff, suggesting that a quadratic dependence on $1/\epsilon$ is required for computationally efficient algorithms. Our main result is a computationally efficient learner with sample complexity $\widetilde{\Theta}(1/(\gamma^2\epsilon^2))$, nearly matching this lower bound. In addition, our algorithm is simple and practical, relying on online SGD on a carefully selected sequence of convex losses.

## 1 Introduction

This work studies the algorithmic task of learning margin halfspaces in the presence of Massart noise (aka bounded label noise) [MN06] with a focus on fine-grained complexity analysis. A halfspace or Linear Threshold Function (LTF) is any Boolean-valued function $h : \mathbb{R}^d \to \{\pm 1\}$ of the form $h(\mathbf{x}) = \text{sign}\,(\mathbf{w} \cdot \mathbf{x} - \theta)$, where $\mathbf{w} \in \mathbb{R}^d$ is the weight vector and $\theta \in \mathbb{R}$ is the threshold. The function $\text{sign} : \mathbb{R} \to \{\pm 1\}$ is defined as $\text{sign}(t) = 1$ if $t \geq 0$ and $\text{sign}(t) = -1$ otherwise. The problem of learning halfspaces with a margin — i.e., under the assumption that no example lies too close to the separating hyperplane — is one of the earliest algorithmic problems studied in machine learning, going back to the Perceptron algorithm [Ros58].

In the realizable PAC model [Val84] (i.e., with clean labels), the sample complexity of learning $\gamma$-margin halfspaces on the unit ball in $\mathbb{R}^d$ is $\Theta(1/(\gamma^2\epsilon))$, where $\epsilon > 0$ is the desired 0-1 error; see, e.g., [SSBD14][1]. Moreover, the Perceptron algorithm is a computationally efficient learner achieving this sample complexity. That is, without label noise, there is a sample-optimal and computationally efficient learner for margin halfspaces.

In this paper, we study the same problem in the Massart noise model that we now define.

**Definition 1.1** (PAC Learning with Massart Noise). Let $D$ be a distribution over $\mathcal{X} \times \{\pm 1\}$, and let $\mathcal{C}$ be a class of Boolean-valued functions over $\mathcal{X}$. We say that $D$ satisfies the $\eta$-Massart noise condition with respect to $\mathcal{C}$, for some $\eta < 1/2$, if there exists a concept $f \in \mathcal{C}$ and an unknown noise function $\eta(\mathbf{x}) : \mathcal{X} \mapsto [0, \eta]$ such that for $(\mathbf{x}, y) \sim D$, the label $y$ satisfies: with probability $1 - \eta(\mathbf{x})$, $y = f(\mathbf{x})$; and $y = -f(\mathbf{x})$ otherwise. Given i.i.d. samples from $D$, the goal of the

---

[1]As is standard, we are assuming that $d = \Omega(1/\gamma^2)$; otherwise, a sample complexity bound of $\widetilde{O}(d/\epsilon)$ follows from standard VC-dimension arguments.

38th Conference on Neural Information Processing Systems (NeurIPS 2024).

learner is to output a hypothesis $h : \mathcal{X} \rightarrow \{\pm 1\}$ such that with high probability the 0-1 error $\mathrm{err}_D(h) \overset{\text{def}}{=} \mathbf{Pr}_{(\mathbf{x},y)\sim D}[h(\mathbf{x}) \neq y]$ is small.

The concept class of halfspaces with a margin is defined as follows.

**Definition 1.2** ($\gamma$-Margin Halfspaces). Let $D$ be a distribution over $\mathbb{S}^{d-1} \times \{\pm 1\}$, where $\mathbb{S}^{d-1}$ is the unit sphere in $\mathbb{R}^d$. Let $\mathbf{w}^* \in \mathbb{S}^{d-1}$ and $\gamma \in (0,1)$. We say that the distribution $D$ satisfies the $\gamma$-margin condition with respect the halfspace $\mathrm{sign}(\mathbf{w}^* \cdot \mathbf{x})^2$, if (i) for $(\mathbf{x}, y) \sim D$, we have that $y = \mathrm{sign}(\mathbf{w}^* \cdot \mathbf{x})$, and (ii) $\mathbf{Pr}_{(\mathbf{x},y)\sim D}[|\mathbf{w}^* \cdot \mathbf{x}| < \gamma] = 0$. The parameter $\gamma$ is called the margin of the halfspace $\mathrm{sign}(\mathbf{w}^* \cdot \mathbf{x})$.

Information-theoretically, the best possible 0-1 error attainable for learning a concept class with Massart noise is $\mathrm{opt} := \mathbf{E}_{\mathbf{x}\sim D_\mathbf{x}}[\eta(\mathbf{x})]$. Since $\eta(\mathbf{x})$ is uniformly bounded above by $\eta$, it follows that $\mathrm{opt} \leq \eta$; also note that it may well be the case that $\mathrm{opt} \ll \eta$. Focusing on the class of $\gamma$-margin halfspaces, it follows from [MN06] that there exists a (computationally inefficient) estimator achieving error $\mathrm{opt} + \epsilon$ with sample complexity $\widetilde{O}(1/((1 - 2\eta)\gamma^2\epsilon))$; and moreover that this sample upper bound is nearly best possible (within a logarithmic factor) for any estimator. (That is, the sample complexity of the Massart learning problem is essentially the same as in the realizable case, as long as $\eta$ is bounded from $1/2$.)

Taking computational considerations into account, the feasibility landscape of the problem changes. Prior work [DK22, NT22, DKMR22] has provided strong evidence that achieving error better than $\eta + \epsilon$ is not possible in polynomial time. Consequently, algorithmic research has been focusing on achieving the qualitatively weaker error guarantee of $\eta + \epsilon$. We note that efficiently obtaining any non-trivial guarantee had remained open since the 80s; see Appendix A.1 for a discussion. The first algorithmic progress for this problem is due to [DGT19], who gave a polynomial-time algorithm achieving error of $\eta + \epsilon$ with sample complexity $\mathrm{poly}(1/\gamma, 1/\epsilon)$. Subsequent work [CKMY20] gave an efficient algorithm with improved sample complexity of $\tilde{O}(1/(\gamma^4\epsilon^3))$. Prior to the current work, this remained the best known sample upper bound for efficient algorithms.

In summary, known computationally efficient algorithms for learning margin halfspaces with Massart noise require significantly more samples—namely, $\tilde{\Omega}(1/(\gamma^4\epsilon^3))$—than the information-theoretic minimum of $\widetilde{\Theta}_\eta(1/(\gamma^2\epsilon))$. It is thus natural to ask whether a polynomial-time algorithm with optimal (or near-optimal, i.e., within logarithmic factors) sample complexity exists. Recall that the answer to this question is affirmative in the realizable setting, where the Perceptron algorithm is optimal. Perhaps surprisingly, recent work [DDK+23a] (see also [DDK+23b]) gave evidence for the existence of inherent *information-computation tradeoffs* in the Massart noise model—in fact, even in the simpler model of Random Classification Noise (RCN) [AL88][3]. Specifically, they showed that any efficient Statistical Query (SQ) algorithm or low-degree polynomial tasks requires $\Omega(1/\epsilon^2)$ samples—a near quadratic blow-up compared to the $\tilde{O}(1/\epsilon)$ information-theoretic upper bound. This discussion serves as the motivation for the following question:

> *What is the optimal* computational sample complexity *of the problem of learning $\gamma$-margin halfspaces with Massart noise?*

By the term "computational sample complexity" above, we mean the sample complexity of polynomial-time algorithms for the problem. Given the fundamental nature of this learning problem, we believe that a fine-grained sample complexity versus computational complexity analysis is interesting on its own merits. *In this work, we develop a computationally efficient algorithm with sample complexity of $\tilde{O}(1/(\gamma^2\epsilon^2))$.* Given the aforementioned information-computation tradeoffs, there is evidence that this upper bound is close to best possible. As a bonus, our algorithm is also simple and practical, relying on online SGD. (In fact, our algorithm runs in sample linear time, excluding a final testing step that slightly increases the runtime.)

## 1.1   Our Result and Techniques

Our main result is the following:

---

[2]We will henceforth assume that the threshold is $\theta = 0$, which is well-known to be no loss of generality.

[3]The RCN model is the special case of Massart noise, where $\eta(\mathbf{x}) = \eta$ for all points $\mathbf{x}$ in the domain.

**Theorem 1.3** (Main Result, Informal). *Let $D$ be a distribution on $\mathbb{S}^{d-1} \times \{\pm 1\}$ that satisfies the $\eta$-Massart noise condition with respect to an unknown $\gamma$-margin halfspace $f(\mathbf{x}) = \text{sign}(\mathbf{w}^* \cdot \mathbf{x})$. There is algorithm that draws $n = \tilde{O}(1/(\epsilon^2\gamma^2))$ samples from $D$, runs in time $\tilde{O}(dn/\epsilon)$, and with probability at least $9/10$ returns a vector $\hat{\mathbf{w}}$ such that $\text{err}_D(\hat{\mathbf{w}}) \leq \eta + \epsilon$.*

The sample upper bound of Theorem 1.3 nearly matches the computational sample complexity of the problem (for SQ algorithms and low-degree polynomial tests), which was shown to be $\Omega(1/(\epsilon^2\gamma) + 1/(\epsilon\gamma^2))$ [MN06, DDK+23a, DDK+23b]. That is, Theorem 1.3 comes close to resolving the fine-grained complexity of this basic task. Moreover, it matches known algorithmic guarantees for the easier case of Random Classification Noise [DDK+23a, KIT+23].

**Independent Work** Independent work [CKST24] obtained a learning algorithm for $\gamma$-margin halfspaces with essentially the same sample and computational complexity as ours.

**Brief Overview of Techniques** Here we provide a brief summary of our approach in tandem with a comparison to prior work. The algorithm of [DGT19] adaptively partitions the space into polyhedral regions and uses a different linear classifier in each region, each achieving error $\eta + \epsilon$ within the corresponding region. Their approach leverages the LeakyReLU loss (see (1)) as a convex proxy to the 0-1 loss. At a high-level, their approach reweights the samples in order to accurately classify a non-trivial fraction of points. [CKMY20] uses the LeakyReLU loss to efficiently identify a region where the value of the loss conditioned on this region is sub-optimal; they then use this procedure as a separation oracle along with online convex optimization (see also [DKTZ20b, DKK+21]) to output a linear classifier with 0-1 error at most $\eta + \epsilon$. Both of these approaches inherently require $\Omega(1/\epsilon^3)$ samples for the following reason: they both need to condition on a region where the probability mass of the distribution can be as small as $\Theta(\epsilon)$. Thus, even estimating the error of the loss would require at least $\Omega(1/\epsilon^2)$ conditional samples. Beyond the dependence on $1/\epsilon$, the sample complexity achieved in these prior works is also suboptimal in the margin parameter $\gamma$; namely, $\Omega(1/\gamma^4)$. This dependence follows from the facts that both of these works require estimating the loss in each iteration within error of at most $\gamma\epsilon$, and that their algorithmic approaches require $\Omega(1/\gamma^2)$ iterations.

To circumvent these issues, novel ideas are required. At a high-level, we design a uniform approach to decrease the "global" error, as opposed to the local error (as was done in prior work). Specifically, we construct a different sequence of convex loss functions, each of which attempts to accurately simulate the 0-1 objective. We note that a similar sequence of loss functions was used in the recent work [DKTZ24] in a related, but significantly different, adversarial online setting. Interestingly, a similar reweighting scheme was used in [CKMY20] for learning general Massart halfspaces. Beyond this similarity, these works have no implications for the sample complexity of our problem. (See Appendix A.2 for a detailed comparison.) Via this approach, we obtain an iterative algorithm which uses only $O_\gamma(1/\epsilon^2)$ samples in order to estimate the loss in each iterative step.

In more detail, note that the 0-1 loss can be written in the form $-\mathbf{E}[y\frac{\mathbf{w}\cdot\mathbf{x}}{|\mathbf{w}\cdot\mathbf{x}|}]$. We convexify this objective by considering, in each step, the loss $\ell(\mathbf{w}, \mathbf{u}) = -\mathbf{E}[y\frac{\mathbf{w}\cdot\mathbf{x}}{|\mathbf{u}\cdot\mathbf{x}|}]$, where $\mathbf{u}$ is independent of $\mathbf{w}$; this loss is convex with respect to $\mathbf{w}$. Observe that $\ell(\mathbf{w}, \mathbf{w})$ is proportional to the zero-one loss of $\mathbf{w}$. Unfortunately, it is possible that no optimal vector $\mathbf{w}^*$ (under 0-1 loss) minimizes $\ell(\mathbf{w}^*, \mathbf{w})$. For this reason, we consider the objective $\ell_\eta(\mathbf{w}, \mathbf{u}) = \mathbf{E}[(\mathbb{1}\{y \neq \text{sign}(\mathbf{w} \cdot \mathbf{x})\} - \eta - \epsilon)|\mathbf{w} \cdot \mathbf{x}|/|\mathbf{u} \cdot \mathbf{x}|]$. This new objective satisfies the following: $\ell_\eta(\mathbf{w}^*, \mathbf{u}) < -\epsilon\gamma$ for any vector $\mathbf{u}$ and any $\mathbf{w}^*$ that minimizes the 0-1 objective; and $\ell_\eta(\mathbf{w}, \mathbf{w}) \geq \epsilon$ as long as $\mathbf{w}$ incurs 0-1 error at least $\eta + \epsilon$. By the convexity of $\ell_\eta(\mathbf{w}, \mathbf{u})$, this allows us to construct a separation oracle. Namely, we draw enough samples so that $\widehat{\ell}_\eta(\mathbf{w}, \mathbf{w}) - \widehat{\ell}_\eta(\mathbf{w}^*, \mathbf{w}) \geq \epsilon/2$, where $\widehat{\ell}$ is the empirical version of the loss. Due to the nature of these objectives, $O_\gamma(1/\epsilon^2)$ samples per iteration suffice for this purpose. This in turn implies that the cutting planes method efficiently finds a near-optimal weight vector after $O(\log(1/\epsilon)/\gamma^2)$ iterations. Overall, this approach leads to an efficient algorithm with sample complexity $\tilde{O}_\gamma(1/\epsilon^2)$. To get the desired sample complexity of $\tilde{O}(1/(\epsilon^2\gamma^2))$, more ideas are needed.

In the previous paragraph, we hid an obstacle that makes the above approach fail. Specifically, it may be possible that, for many points $\mathbf{x}$, the value of $|\mathbf{u} \cdot \mathbf{x}|$ is arbitrarily small. To fix this issue, we consider a clipped reweighting as follows: $\ell'_\eta(\mathbf{w}, \mathbf{u}) = \mathbf{E}[(\mathbb{1}\{y \neq \text{sign}(\mathbf{w} \cdot \mathbf{x})\} - \eta - \epsilon)\frac{|\mathbf{w}\cdot\mathbf{x}|}{\max(|\mathbf{u}\cdot\mathbf{x}|,\gamma)}]$. This clipping step is not a problem for us, because the target halfspace $\text{sign}(\mathbf{w}^* \cdot \mathbf{x})$ was assumed to have margin $\gamma$. This guarantees that the difference between the expected (over $y$) pointwise losses at $(\mathbf{w}, \mathbf{w})$ and $(\mathbf{w}^*, \mathbf{w})$ is at least $\epsilon$ on the points $\mathbf{x}$ where $|\mathbf{u} \cdot \mathbf{x}| \leq \gamma$. Indeed, when this is the case, then $|\mathbf{w}^* \cdot \mathbf{x}|/|\mathbf{u} \cdot \mathbf{x}| \geq 1$. Overall, this suffices to guarantee that $\ell'_\eta(\mathbf{w}, \mathbf{w}) - \ell'_\eta(\mathbf{w}^*, \mathbf{w}) \geq \epsilon$.

## 1.2 Notation

For $n \in \mathbb{Z}_+$, let $[n] \stackrel{\text{def}}{=} \{1, \ldots, n\}$. We use small boldface characters for vectors. For $\mathbf{x} \in \mathbb{R}^d$ and $i \in [d]$, $\mathbf{x}_i$ denotes the $i$-th coordinate of $\mathbf{x}$, and $\|\mathbf{x}\|_2 \stackrel{\text{def}}{=} (\sum_{i=1}^d \mathbf{x}_i^2)^{1/2}$ denotes the $\ell_2$-norm of $\mathbf{x}$. We will use $\mathbf{x} \cdot \mathbf{y}$ for the inner product of $\mathbf{x}, \mathbf{y} \in \mathbb{R}^d$. For a subset $S \subseteq \mathbb{R}^d$, we define the $\mathrm{proj}_S$ operator that maps a point $\mathbf{x} \in \mathbb{R}^d$ to the closest point in the set $S$. For $a, b \in \mathbb{R}$, we denote $W(a, b) \stackrel{\text{def}}{=} 1/\max(a, b)$. We will use $\mathbb{1}_A$ to denote the characteristic function of the set $A$, i.e., $\mathbb{1}\{\mathbf{x} \in A\} = 1$ if $\mathbf{x} \in A$, and $\mathbb{1}\{\mathbf{x} \in A\} = 0$ if $\mathbf{x} \notin A$. For $A, B \in \mathbb{R}$, we write $A \gtrsim B$ (resp. $A \lesssim B$) to denote that there exists a universal constant $C > 0$, such that $A \geq CB$ (resp. $A \leq CB$).

We use $\mathbf{E}_{x \sim D}[x]$ for the expectation of the random variable $x$ with respect to the distribution $D$ and $\mathbf{Pr}[\mathcal{E}]$ for the probability of event $\mathcal{E}$. For simplicity, we may omit the distribution when it is clear from the context. For $(\mathbf{x}, y) \sim D$, we use $D_{\mathbf{x}}$ for the marginal distribution of $\mathbf{x}$ and $D_y(\mathbf{x})$ for the distribution of $y$ conditioned on $\mathbf{x}$. We use $\widehat{D}_N$ to denote the empirical distribution obtained by drawing $N$ i.i.d. samples from $D$. We use $\mathrm{err}_D(\mathbf{w})$ to denote the 0-1 error of the halfspace defined by the weight vector $\mathbf{w}$ with respect to the distribution $D$, i.e., $\mathrm{err}_D(\mathbf{w}) \stackrel{\text{def}}{=} \mathbf{Pr}_{(\mathbf{x}, y) \sim D}[\mathrm{sign}(\mathbf{w} \cdot \mathbf{x}) \neq y]$. We will use $\mathrm{err}(\mathbf{w}, \mathbf{x})$ for the 0-1 error of $\mathrm{sign}(\mathbf{w} \cdot \mathbf{x})$ conditioned on $\mathbf{x}$, i.e., $\mathrm{err}(\mathbf{w}, \mathbf{x}) := \mathbf{Pr}_{y \sim D_y(\mathbf{x})}[\mathrm{sign}(\mathbf{w} \cdot \mathbf{x}) \neq y]$. Note that $\mathrm{err}_D(\mathbf{w}) = \mathbf{E}_{\mathbf{x} \sim D_{\mathbf{x}}}[\mathrm{err}(\mathbf{w}, \mathbf{x})]$. If $D$ satisfies the $\eta$-Massart noise condition with respect to the halfspace $\mathrm{sign}(\mathbf{w} \cdot \mathbf{x})$, then $\mathrm{err}(\mathbf{w}, \mathbf{x}) = \eta(\mathbf{x}) \mathbb{1}\{\mathrm{sign}(\mathbf{w} \cdot \mathbf{x}) = \mathrm{sign}(\mathbf{w}^* \cdot \mathbf{x})\} + (1 - \eta(\mathbf{x})) \mathbb{1}\{\mathrm{sign}(\mathbf{w} \cdot \mathbf{x}) \neq \mathrm{sign}(\mathbf{w}^* \cdot \mathbf{x})\}$.

## 2 Our Algorithm and its Analysis: Proof of Theorem 1.3

In this section, we prove our main result. Algorithm 1 efficiently learns the class of margin halfspaces on the unit ball, in the presence of Massart noise, with sample complexity nearly matching the information-computation limit. Additionally, its runtime is linear in the sample size, excluding a final testing step to select the best hypothesis.

At a high-level, our algorithm leverages a carefully selected convex loss (or, more precisely, a sequence of convex losses) — serving as a proxy to the 0-1 error. A common loss function, introduced in this context by [DGT19] and leveraged in [DGT19, CKMY20], is the LeakyReLU function. This is the univariate function $\mathrm{LeakyReLU}_\lambda(t) = (1 - \lambda) \mathbb{1}\{t \geq 0\} t + \lambda \mathbb{1}\{t < 0\} t$, where $\lambda \in (0, 1)$ is the leakage parameter (that needs to be selected carefully). Roughly speaking, the convex function $\ell_\lambda(\mathbf{w}, \mathbf{x}, y) = \mathrm{LeakyReLU}_\lambda(-y(\mathbf{w} \cdot \mathbf{x}))$ can be viewed as a reasonable proxy to the 0-1 loss of the halfspace $\mathrm{sign}(\mathbf{w} \cdot \mathbf{x})$ on the point $(\mathbf{x}, y)$. To see this, note that (see, e.g., Claim C.1)

$$\ell_\lambda(\mathbf{w}, \mathbf{x}, y) = (\mathbb{1}\{\mathrm{sign}(\mathbf{w} \cdot \mathbf{x}) \neq y\} - \lambda)|\mathbf{w} \cdot \mathbf{x}| . \tag{1}$$

Observe that a point $\mathbf{x}$ that is classified correctly by the halfspace $\mathrm{sign}(\mathbf{w} \cdot \mathbf{x})$ will satisfy

$$\left(\mathbf{E}_{y \sim D_y(\mathbf{x})}[\mathbb{1}\{\mathrm{sign}(\mathbf{w} \cdot \mathbf{x}) \neq y\}] - \lambda\right)|\mathbf{w} \cdot \mathbf{x}| = (\eta(\mathbf{x}) - \lambda)|\mathbf{w} \cdot \mathbf{x}|$$

which is non-positive for $\lambda \geq \eta(\mathbf{x})$. Since the only guarantee we have is that $\eta(\mathbf{x}) \leq \eta$, this suggests that we need to select $\lambda \geq \eta$. It turns out that $\lambda := \eta$ is the optimal choice. We fix the choice of $\lambda := \eta$ throughout. On the other hand, if (the halfspace defined by) $\mathbf{w}$ misclassifies the point $\mathbf{x}$, this term becomes non-negative.

The factor $|\mathbf{w} \cdot \mathbf{x}|$ in Equation (1) reweights the 0-1 error so that points $\mathbf{x}$ for which $|\mathbf{w} \cdot \mathbf{x}|$ is sufficiently large (i.e., close to 1) have to be classified correctly by a minimizer of $\mathbf{E}_{(\mathbf{x}, y) \sim D}[\ell_\lambda(\mathbf{w}, \mathbf{x}, y)]$. On the other hand, points closer to the separating hyperplane defined by $\mathbf{w}$, or points where $\eta(\mathbf{x})$ is close to $\lambda = \eta$, are not guaranteed to be classified correctly by the minimizer of this loss. We leverage this insight to construct a sequence of loss functions that reweight the points so that, to minimize the regret, we need to classify a large fraction of points; this leads to the desired error of $\eta + \epsilon$ with near-optimal sample complexity.

We now provide some intuition justifying our choice of surrogate loss functions. Observe that if we instead could minimize the function

$$\mathbf{E}_{(\mathbf{x}, y) \sim D}[\ell_\lambda(\mathbf{w}, \mathbf{x}, y)/|\mathbf{w} \cdot \mathbf{x}|] = \mathbf{E}_{(\mathbf{x}, y) \sim D}[(\mathbb{1}\{\mathrm{sign}(\mathbf{w} \cdot \mathbf{x}) \neq y\} - \lambda)] , \tag{2}$$

with respect to $\mathbf{w}$, we would obtain a halfspace with minimum 0-1 error; unfortunately, this reweighted loss is just a shift of the 0-1 loss, hence non-convex. To fix this issue, instead of reweighting by

$1/|\mathbf{w} \cdot \mathbf{x}|$, we will reweight by $W(\mathbf{v} \cdot \mathbf{x}, \gamma) \stackrel{\text{def}}{=} 1/\max(|\mathbf{v} \cdot \mathbf{x}|, \gamma)$, where $\gamma$ is the margin parameter and $\mathbf{v}$ is an appropriately chosen vector that is independent of $\mathbf{w}$. The new loss is defined as follows:

$$\mathcal{L}_{\lambda,\mathbf{v}}(\mathbf{w}) \stackrel{\text{def}}{=} \mathop{\mathbf{E}}_{(\mathbf{x},y)\sim D}[\ell_\lambda(\mathbf{w}, \mathbf{x}, y)W(\mathbf{v} \cdot \mathbf{x}, \gamma/2)] \,, \tag{3}$$

where for technical reasons we use $\gamma/2$ instead of $\gamma$ in the maximum.

Since the parameter $\mathbf{v}$ is independent of $\mathbf{w}$, the loss $\mathcal{L}_{\lambda,\mathbf{v}}(\mathbf{w})$ remains convex in $\mathbf{w}$. At the same time, by carefully choosing $\mathbf{v}$, we can accurately simulate the non-convex 0-1 loss. Note that our reweighting term is a maximum over two terms. The reason for this choice is that, for some points $\mathbf{x}$, the quantity $|\mathbf{v} \cdot \mathbf{x}|$ can be arbitrarily small; taking the maximum avoids the loss becoming very large. In particular, the loss $\mathcal{L}_{\lambda,\mathbf{v}}(\mathbf{w})$ will be guaranteed to remain in a bounded length interval.

Our algorithm proceeds in a sequence of iterations. In the $(t+1)$-st iteration, it sets $\mathbf{v}$ to be $\mathbf{w}^t$, where $\mathbf{w}^t$ is the weight vector of step $t$. This choice attempts to simulate the 0-1 error at $\mathbf{w}^t$, as is suggested by Equation (2). Assume for simplicity that our current hypothesis is the halfspace defined by $\mathbf{w}$ and is such that $\mathbf{E}_{\mathbf{x}\sim D_\mathbf{x}}[\mathbb{1}\{|\mathbf{w} \cdot \mathbf{x}| \leq \gamma/2\}] = 0$. Note this implies that $W(\mathbf{w} \cdot \mathbf{x}, \gamma/2) = 1/|\mathbf{w} \cdot \mathbf{x}|$. By combining Equations (2) and (3), we get that $\mathcal{L}_{\lambda,\mathbf{w}}(\mathbf{w}) = \mathrm{err}_D(\mathbf{w}) - \lambda$; note that as long as $\mathrm{err}_D(\mathbf{w}) \geq \lambda + \epsilon$, we have that $\mathcal{L}_{\lambda,\mathbf{w}}(\mathbf{w}) \geq \epsilon$. On the other hand, the optimal halfspace $\mathbf{w}^*$ achieves a non-positive loss; from Equations (1) and (2), we have that

$$\begin{aligned}
\mathcal{L}_{\lambda,\mathbf{w}}(\mathbf{w}^*) &= \mathop{\mathbf{E}}_{(\mathbf{x},y)\sim D}[(\mathbb{1}\{\mathrm{sign}(\mathbf{w}^* \cdot \mathbf{x}) \neq y\} - \lambda)|\mathbf{w}^* \cdot \mathbf{x}|W(\mathbf{w} \cdot \mathbf{x}, \gamma/2)] \\
&= \mathop{\mathbf{E}}_{\mathbf{x}\sim D_\mathbf{x}}[(\eta(\mathbf{x}) - \lambda)|\mathbf{w}^* \cdot \mathbf{x}|W(\mathbf{w} \cdot \mathbf{x}, \gamma/2)] \leq 0 \,,
\end{aligned}$$

where the inequality follows from the fact that $\eta(\mathbf{x}) \leq \eta$. Recalling that $\mathcal{L}_{\lambda,\mathbf{v}}(\mathbf{w})$ is convex, if we run an Online Convex Optimization (OCO) algorithm, after $T$ steps we are guaranteed to find a vector $\mathbf{w}$ such that $\mathcal{L}_{\lambda,\mathbf{w}}(\mathbf{w}) - \mathcal{L}_{\lambda,\mathbf{w}}(\mathbf{w}^*) \leq O(1/\sqrt{T})$. For $T = O(1/\epsilon^2)$, this gives that $\mathcal{L}_{\lambda,\mathbf{w}}(\mathbf{w}) < \epsilon/2$; and therefore we would have $\mathrm{err}_D(\mathbf{w}) < \lambda + \epsilon$. We provide an approach using this idea and the cutting planes algorithm in Appendix B that achieves sample complexity $\widetilde{O}(1/(\epsilon^2\gamma^4))$.

Our algorithm and its analysis work only with the gradient of $\mathcal{L}_{\lambda,\mathbf{v}}(\mathbf{w})$. The key novelty is the analysis of the sample complexity. The gradient of $\ell_\lambda(\mathbf{w}, \mathbf{x}, y)W(\mathbf{v} \cdot \mathbf{x}, \gamma)$ with respect to $\mathbf{w}$ has the following explicit form:

$$\mathbf{g}_{\lambda,\gamma}(\mathbf{w}, \mathbf{v}, \mathbf{x}, y) \stackrel{\text{def}}{=} ((1 - 2\lambda)\mathrm{sign}(\mathbf{w} \cdot \mathbf{x}) - y)W(\mathbf{v} \cdot \mathbf{x}, \gamma)\mathbf{x} = \frac{((1 - 2\lambda)\mathrm{sign}(\mathbf{w} \cdot \mathbf{x}) - y)}{\max(|\mathbf{v} \cdot \mathbf{x}|, \gamma)}\mathbf{x} \,.$$

Furthermore, we denote by $\mathbf{G}_D(\mathbf{w}, \mathbf{v}, \eta, \gamma) = \mathbf{E}_{(\mathbf{x},y)\sim D}[\mathbf{g}_{\eta,\gamma}(\mathbf{w}, \mathbf{v}, \mathbf{x}, y)]$.

Before describing our algorithm and proving Theorem 2.1, we simplify our notation. We will omit the parameters $\eta, \gamma$ from the function input (as they are fixed throughout). Therefore, we use $\mathbf{G}_{\widehat{D}_N^t}(\mathbf{w}, \mathbf{v}) \equiv \mathbf{G}_{\widehat{D}_N^t}(\mathbf{w}, \mathbf{v}, \eta, \gamma)$ and $\mathbf{g}(\mathbf{w}, \mathbf{v}, \mathbf{x}, y) \equiv \mathbf{g}_{\eta,\gamma/2}(\mathbf{w}, \mathbf{v}, \mathbf{x}, y)$.

Our algorithm is described in pseudocode below.

Algorithm 1 employs online SGD applied to a sequence of convex loss functions. We show that, after a certain number of iterations, the algorithm will find a weight vector achieving 0-1 error at most $\eta + \epsilon$. Since the desired vector may not be the last iterate, in the end, our algorithm returns the halfspace that achieves the smallest empirical 0-1 error.

We establish the following result, which implies Theorem 1.3.

**Theorem 2.1** (Main Result). *Let $D$ be a distribution on $\mathbb{S}^{d-1} \times \{\pm 1\}$ satisfying the $\eta$-Massart noise condition with respect to the $\gamma$-margin halfspace $f(\mathbf{x}) = \mathrm{sign}(\mathbf{w}^* \cdot \mathbf{x})$. Given $N = \Theta(\log(1/(\gamma\delta))/\epsilon(1 - 2\eta))$ and $T = \Theta(\log(1/\delta)/(\epsilon^2\gamma^2))$, Algorithm 1 returns a vector $\hat{\mathbf{w}}$ such that $\mathrm{err}_D(\hat{\mathbf{w}}) \leq \eta + \epsilon$ with probability at least $1 - \delta$. The algorithm draws $n = O(N + T)$ samples from $D$ and runs in $O(dNT)$ time.*

The rest of this section is devoted to the proof of Theorem 2.1.

Our algorithm sets $\mathbf{v} = \mathbf{w}^t$ in each round, therefore for the rest of the section we proceed by setting $\mathbf{v} = \mathbf{w}$ as arguments of $\mathbf{g}$ and $\mathbf{G}$.

> **Input:** Sample access to a distribution $D$ supported in $\mathbb{S}^{d-1} \times \{\pm 1\}$ corrupted with $\eta$-Massart noise with respect to a halfspace $\text{sign}(\mathbf{w}^* \cdot \mathbf{x})$ that satisfies the $\gamma$-margin condition; parameters $\epsilon, \delta \in (0,1)$, and $N, T \in \mathbb{Z}_+$.
> **Output:** Weight vector $\hat{\mathbf{w}}$ such that $\text{err}_D(\hat{\mathbf{w}}) \leq \eta + \epsilon$ with probability at least $1 - \delta$.
>
> 1. Let $c > 0$ be a sufficiently small universal constant.
> 2. $t \leftarrow 0$, $\mathbf{w}^0 \leftarrow \mathbf{e}_1 = (1, 0, \ldots, 0)$, and $T = (1/c) \log(1/\delta)/(\epsilon^2 \gamma^2)$.
> 3. While $t \leq T$ do
>     (a) Draw $(\mathbf{x}^{(t)}, y^{(t)})$ sample from $D$.
>     (b) Set $\lambda_t \leftarrow c\gamma^2 \epsilon$.
>     (c) Update $\mathbf{w}^t$ as follows:          ▷ Update and project in the unit ball
>
>     $$\mathbf{v}^{t+1} \leftarrow \mathbf{w}^t - \lambda_t \mathbf{g}(\mathbf{w}^t, \mathbf{w}^t, \mathbf{x}^{(t)}, y^{(t)}) \qquad \mathbf{w}^{t+1} \leftarrow \frac{\mathbf{v}^{t+1}}{\max(\|\mathbf{v}^{t+1}\|_2, 1)}$$
>
>     (d) $t \leftarrow t + 1$.
> 4. Draw $N$ samples from $D$ and construct the empirical distribution $\widehat{D}_N$.
> 5. Return $\hat{\mathbf{w}} = \text{argmin}_{t \in [T+1]} \text{err}_{\widehat{D}_N}(\mathbf{w}^t)$.

**Algorithm 1:** Learning Margin Halfspaces with Massart Noise

We decompose the stochastic gradient $\mathbf{g}(\mathbf{w}, \mathbf{w}, \mathbf{x}, y)$ into two parts: $\mathbf{g}(\mathbf{w}, \mathbf{w}, \mathbf{x}, y) = \mathbf{g}^1(\mathbf{w}, \mathbf{x}) + \mathbf{g}^2(\mathbf{w}, \mathbf{x}, y)$, where

$$\mathbf{g}^1(\mathbf{w}, \mathbf{x}) = \left( (1 - 2\eta)\text{sign}(\mathbf{w} \cdot \mathbf{x}) - \mathop{\mathbf{E}}_{y \sim D_y(\mathbf{x})}[y] \right) W(\mathbf{w} \cdot \mathbf{x}, \gamma/2)\mathbf{x}$$

and

$$\mathbf{g}^2(\mathbf{w}, \mathbf{x}, y) = \left( \mathop{\mathbf{E}}_{y \sim D_y(\mathbf{x})}[y] - y \right) W(\mathbf{w} \cdot \mathbf{x}, \gamma/2)\mathbf{x} \ .$$

We also use $\mathbf{G}^1_{\widehat{D}_N}(\mathbf{w})$ and $\mathbf{G}^2_{\widehat{D}_N}(\mathbf{w})$ for the same decomposition after taking the empirical expectation, i.e., $\mathbf{G}^1_{\widehat{D}_N}(\mathbf{w}) = \mathbf{E}_{\mathbf{x} \sim (\widehat{D}_\mathbf{x})_N}[\mathbf{g}^1(\mathbf{w}, \mathbf{x})]$ and $\mathbf{G}^2_{\widehat{D}_N}(\mathbf{w}) = \mathbf{E}_{(\mathbf{x},y) \sim \widehat{D}_N}[\mathbf{g}^2(\mathbf{w}, \mathbf{x}, y)]$.

This serves to decompose the gradient into two parts: one containing the population expectation over the random variable $y$, and the other containing the error between the empirical estimation of $y$ and the population version of $y$. The vector $\mathbf{G}^1_{\widehat{D}_N}(\mathbf{w})$ contains the direction that will decrease the distance between $\mathbf{w}$ and $\mathbf{w}^*$, while $\mathbf{G}^2_{\widehat{D}_N}(\mathbf{w})$ contains the estimation error. To see this, observe that if we take the population expectation of $\mathbf{g}^2(\mathbf{w}, \mathbf{x}, y)$, we will have:

$$\mathop{\mathbf{E}}_{(\mathbf{x},y) \sim D}[\mathbf{g}^2(\mathbf{w}, \mathbf{x}, y)] = \mathop{\mathbf{E}}_{\mathbf{x} \sim D_\mathbf{x}} \left[ \left( (1 - 2\eta(\mathbf{x}))\text{sign}(\mathbf{w}^* \cdot \mathbf{x}) - \mathop{\mathbf{E}}_{y \sim D_y(\mathbf{x})}[y] \right) W(\mathbf{w} \cdot \mathbf{x}, \gamma/2)\mathbf{x} \right] = 0 \ ,$$

where we used that $\mathbf{E}_{y \sim D_y(\mathbf{x})}[y] = (1 - 2\eta(\mathbf{x}))\text{sign}(\mathbf{w}^* \cdot \mathbf{x})$.

We start by bounding the contribution of $\mathbf{G}^1_{\widehat{D}_N}(\mathbf{w})$ in the direction $\mathbf{w} - \mathbf{w}^*$. We show that if instead of the corrupted label $y$ at the point $\mathbf{x}$, we had access to $\mathbf{E}_{y \sim D_y(\mathbf{x})}[y] = (1 - 2\eta(\mathbf{x}))\text{sign}(\mathbf{w}^* \cdot \mathbf{x})$, then the gradient has a large component in the direction of $\mathbf{w} - \mathbf{w}^*$. This effectively implies that $\mathbf{G}^1_{\widehat{D}_N}(\mathbf{w})$ can be used as a separation oracle, separating all the halfspaces with 0-1 error more than $\eta + \epsilon$ from the ones with smaller error.

**Lemma 2.2** (Structural Lemma). *Let $N \in \mathbb{Z}_+$ and let $D$ be a distribution on $\mathbb{S}^{d-1} \times \{\pm 1\}$ satisfying the $\eta$-Massart condition with respect to the optimal classifier $f(\mathbf{x}) = \text{sign}(\mathbf{w}^* \cdot \mathbf{x})$. Let $\mathbf{w} \in \mathbb{R}^d$ be such that $\|\mathbf{w}\|_2 \leq 1$ and let $\{\mathbf{x}^{(i)}\}_{i=1}^N$ be a multiset of $N$ i.i.d. samples from $D_\mathbf{x}$. Then, it holds $\mathbf{G}^1_{\widehat{D}_N}(\mathbf{w}) \cdot (\mathbf{w} - \mathbf{w}^*) \geq 2(\text{err}_{\widehat{D}_N}(\mathbf{w}) - \eta)$ , where $\widehat{D}_N$ is the corresponding empirical distribution.*

*Proof.* We partition $\mathbb{R}^d$ into two subsets $R_1, R_2$ as follows: $R_1$ contains the points that lie sufficiently far away from the separating hyperplane $\mathbf{w} \cdot \mathbf{x} = 0$, i.e., $R_1 \stackrel{\text{def}}{=} \{\mathbf{x} \in \mathbb{R}^d : |\mathbf{w} \cdot \mathbf{x}| \geq \gamma/2\}$. $R_2$ contains the remaining points, i.e., $R_2 \stackrel{\text{def}}{=} \{\mathbf{x} \in \mathbb{R}^d : |\mathbf{w} \cdot \mathbf{x}| < \gamma/2\}$.

We first show that for any $\mathbf{x} \in R_1$, the vector $\mathbf{g}^1(\mathbf{w}, \mathbf{x})$ has a large component parallel to the direction $\mathbf{w} - \mathbf{w}^*$. The proof of the claim below can be found in Appendix C.

**Claim 2.3.** *For any $\mathbf{x}^{(i)} \in R_1$, we have that $\mathbf{g}^1(\mathbf{w}, \mathbf{x}^{(i)}) \cdot (\mathbf{w} - \mathbf{w}^*) \geq 2(\mathrm{err}(\mathbf{w}, \mathbf{x}^{(i)}) - \eta)$ .*

It remains to show that the same holds for all the points in $R_2$. The proof of the claim below can be found in Appendix C.

**Claim 2.4.** *For any $\mathbf{x}^{(i)} \in R_2$, we have that $\mathbf{g}^1(\mathbf{w}, \mathbf{x}^{(i)}) \cdot (\mathbf{w} - \mathbf{w}^*) \geq 2(\mathrm{err}(\mathbf{w}, \mathbf{x}^{(i)}) - \eta)$ .*

Applying Claim 2.3 and Claim 2.4 for each sample in the set $\{\mathbf{x}^{(i)}\}_{i=1}^N$, we get that

$$\frac{1}{N} \sum_{i=1}^N \mathbf{g}^1(\mathbf{w}, \mathbf{x}^{(i)}) \cdot (\mathbf{w} - \mathbf{w}^*) \geq \frac{2}{N} \sum_{i=1}^N (\mathrm{err}(\mathbf{w}, \mathbf{x}^{(i)}) - \eta) .$$

This completes the proof of Lemma 2.2. $\qquad\qquad\qquad\qquad\qquad\qquad\qquad\qquad\qquad$ $\square$

By Lemma 2.2, the gradient points towards the direction $\mathbf{w}^t - \mathbf{w}^*$, in the $t$-th iteration. This means that, in fact, the gradient is a subgradient of the potential loss $\Phi(\mathbf{w}) = \|\mathbf{w} - \mathbf{w}^*\|_2^2$. This allows us to show convergence, even though it is generally not possible in a sequence of loss functions in the stochastic setting. We are now ready to prove our main result.

*Proof of Theorem 2.1.* Let $T$ be the maximum number of iterations of Algorithm 1. Denote by $\mathcal{Z}^t := \{(\mathbf{x}^{(t)}, y^{(t)})\}$ the i.i.d. sample drawn from $D$ in the $t$-th iteration, $t \in [T]$. Furthermore, let $\mathcal{F}_1, \ldots, \mathcal{F}_T$ be the filtration with respect to the $\sigma$-algebra generated by $\mathcal{Z}^1, \ldots, \mathcal{Z}^T$. We denote by $H_t$ the event that $\mathrm{err}_D(\mathbf{w}^t) \geq \eta + \epsilon$.

Recall that Algorithm 1 uses the following update rule (see Step (3c)):

$$\mathbf{w}^{t+1} = \mathrm{proj}_{\{\mathbf{w} \in \mathbb{R}^d : \|\mathbf{w}\|_2 \leq 1\}}(\mathbf{w}^t - \lambda_t \mathbf{g}(\mathbf{w}^t, \mathbf{w}^t, \mathbf{x}^{(t)}, y^{(t)})) ,$$

with $\lambda_t = c\gamma^2 \epsilon$ , for some sufficiently small absolute constant $c > 0$.

We begin by bounding from above the distance between $\mathbf{w}^{t+1}$ and $\mathbf{w}^*$ from the previous distance between $\mathbf{w}^t$ and $\mathbf{w}^*$. We have that

$$\begin{aligned}
\|\mathbf{w}^{t+1} - \mathbf{w}^*\|_2^2 &= \|\mathrm{proj}_{\{\mathbf{w} \in \mathbb{R}^d : \|\mathbf{w}\|_2 \leq 1\}}(\mathbf{w}^t - \lambda_t \mathbf{g}(\mathbf{w}^t, \mathbf{w}^t, \mathbf{x}^{(t)}, y^{(t)}) - \mathbf{w}^*\|_2^2 \\
&\leq \|\mathbf{w}^t - \lambda_t \mathbf{g}(\mathbf{w}^t, \mathbf{w}^t, \mathbf{x}^{(t)}, y^{(t)}) - \mathbf{w}^*\|_2^2 \\
&= \|\mathbf{w}^t - \mathbf{w}^*\|_2^2 - 2\lambda_t \mathbf{g}(\mathbf{w}^t, \mathbf{w}^t, \mathbf{x}^{(t)}, y^{(t)}) \cdot (\mathbf{w}^t - \mathbf{w}^*) + \lambda_t^2 \|\mathbf{g}(\mathbf{w}^t, \mathbf{w}^t, \mathbf{x}^{(t)}, y^{(t)})\|_2^2 ,
\end{aligned}$$

$$(4)$$

where in the first inequality we used the projection inequality, i.e., $\|\mathrm{proj}_B(\mathbf{v}) - \mathrm{proj}_B(\mathbf{u})\|_2 \leq \|\mathbf{v} - \mathbf{u}\|_2$ for any set $B$. We will decouple the mean of the random variable $\mathbf{g}(\mathbf{w}^t, \mathbf{w}^t, \mathbf{x}, y)$ and make it zero-mean.

To simplify the notation, we denote by $\xi_t := \left(\mathbf{g}(\mathbf{w}^t, \mathbf{w}^t, \mathbf{x}^{(t)}, y^{(t)}) - \mathbf{G}_D^1(\mathbf{w}^t)\right) \cdot (\mathbf{w}^t - \mathbf{w}^*)$ and note that $\xi_t$ is a zero-mean random variable over the sample $(\mathbf{x}^{(t)}, y^{(t)})$. Adding and subtracting $\mathbf{G}_D^1(\mathbf{w}^t)$ onto Inequality (4) a we get that

$$\|\mathbf{w}^{t+1} - \mathbf{w}^*\|_2^2 \leq \|\mathbf{w}^t - \mathbf{w}^*\|_2^2 \underbrace{-2\lambda_t \mathbf{G}_D^1(\mathbf{w}^t) \cdot (\mathbf{w}^t - \mathbf{w}^*) + \lambda_t^2 \|\mathbf{g}(\mathbf{w}^t, \mathbf{w}^t, \mathbf{x}^{(t)}, y^{(t)})\|_2^2}_{I} \underbrace{-2\lambda_t \xi_t}_{\widehat{V}_t} .$$

$$(5)$$

We now outline the main steps of our analysis. Instead of accurately estimating the gradients in each round, we denote by $\widehat{V}_t$ the estimation error from which we bound above their sum. We first add and

subtract the population gradient to obtain the $I$ term, which is the decreasing direction. In this way, we decouple the expected decrease and the error of the approximation (see Claim 2.5). After that, we bound the contribution of the estimation error in Lemma 2.8. Observe that $\widehat{V}_t$ is a random variable that corresponds to the estimation error of the gradient. We will argue that with high probability the contribution of $\sum_{t=1}^{T} \widehat{V}_t$ is bounded; therefore, our algorithm will converge to an accurate solution.

Lemma 2.2 shows that $\mathbf{G}^1_{\widehat{D}^t_N}(\mathbf{w}^t)$ (and therefore the same holds for $\mathbf{G}^1_D(\mathbf{w}^t)$) contains substantial contribution towards to the direction $\mathbf{w}^t - \mathbf{w}^*$, depending of the current error. We show that our choice of step size guarantees a decreasing direction. To this end, we prove the following:

**Claim 2.5.** *Assume that the event $H_t$ happens, i.e., $\mathrm{err}_D(\mathbf{w}^t) \geq \eta + \epsilon$. If $\lambda_t \leq \gamma^2 \epsilon/8$, then $I \leq -\lambda_t(\mathrm{err}_D(\mathbf{w}^t) - \eta)$.*

*Proof of Claim 2.5.* Recall that $I = -2\lambda_t \mathbf{G}^1_D(\mathbf{w}^t) \cdot (\mathbf{w}^t - \mathbf{w}^*) + \lambda_t^2 \|\mathbf{g}(\mathbf{w}^t, \mathbf{w}^t, \mathbf{x}^{(t)}, y^{(t)})\|_2^2$. By Lemma 2.2, we get that $\mathbf{G}^1_{\widehat{D}_N}(\mathbf{w}^t) \cdot (\mathbf{w}^t - \mathbf{w}^*) \geq 2(\mathrm{err}_{\widehat{D}_N}(\mathbf{w}^t) - \eta)$; hence, by taking expectations over the samples, we also have $\mathbf{G}^1_D(\mathbf{w}^t) \cdot (\mathbf{w}^t - \mathbf{w}^*) \geq 2(\mathrm{err}_D(\mathbf{w}^t) - \eta)$. Furthermore, we have that $\|\mathbf{g}(\mathbf{w}^t, \mathbf{w}^t, \mathbf{x}^{(t)}, y^{(t)})\|_2^2 \leq 8/\gamma^2$. Hence, $I \leq -2\lambda_t(\mathrm{err}_D(\mathbf{w}^t) - \eta) + 8(\lambda_t^2/\gamma^2)$ . The claim follows by noting that if $\lambda_t \leq \gamma^2 \epsilon/8$, then $-\lambda_t(\mathrm{err}_D(\mathbf{w}^t) - \eta) + 8(\lambda_t^2/\gamma^2) \leq 0$. Therefore, we obtain

$$I \leq -\lambda_t(\mathrm{err}_D(\mathbf{w}^t) - \eta) .$$

This completes the proof of Claim 2.5. $\qquad\square$

Therefore, our choice of parameters guarantees that $\lambda_t \leq \gamma^2 \epsilon/8$. Using Claim 2.5 onto Inequality (5), we have that

$$\|\mathbf{w}^{t+1} - \mathbf{w}^*\|_2^2 \leq \|\mathbf{w}^t - \mathbf{w}^*\|_2^2 - \lambda_t(\mathrm{err}_D(\mathbf{w}^t) - \eta) + \widehat{V}_t . \tag{6}$$

Using Claim 2.5 and Inequality (6), we have that

$$\|\mathbf{w}^{T+1} - \mathbf{w}^*\|_2^2 \leq \|\mathbf{w}^T - \mathbf{w}^*\|_2^2 - \lambda_T(\mathrm{err}_D(\mathbf{w}^T) - \eta) + \widehat{V}_T$$

$$\leq \|\mathbf{w}^0 - \mathbf{w}^*\|_2^2 - \sum_{t=0}^{T} \lambda_t(\mathrm{err}_D(\mathbf{w}^t) - \eta) + \sum_{t=0}^{T} \widehat{V}_t . \tag{7}$$

To complete the proof of Theorem 2.1, we need to bound the estimation error that corresponds to the random variable $\widehat{V}_t$. We show that $\widehat{V}_t$ does not increase the error by a lot. Recall that $\widehat{V}_t = -2\lambda_t \xi_t$ .

Before proceeding, we provide some basic background on subgaussian random variables.

**Definition 2.6** (Subgaussian Random Variable). For $\sigma > 0$, a zero-mean random variable $X \in \mathbb{R}$ is called $\sigma$-subgaussian, if for any $\lambda \in \mathbb{R}$ it holds $\log(\mathbf{E}[\exp(\lambda X)]) \leq \lambda^2 \sigma^2$ .

Note that any zero-mean bounded random variable is subgaussian. Specifically, we have the following:

**Fact 2.7** (Hoeffding's lemma, see, e.g., [Ver18]). *Let $X \in \mathbb{R}$ be a zero-mean random variable such that $|X| \leq \sigma$ for some $\sigma > 0$. Then $X$ is $C\sigma$-subgaussian, where $C > 0$ is a universal constant.*

Equipped with the above context, we show the following:

**Lemma 2.8.** *With probability at least $1 - \delta$ over the random samples, it holds that $\sum_{t=0}^{T} \widehat{V}_t \leq C\gamma^2 \epsilon^2 T + \log(1/\delta)$, where $C > 0$ is an absolute constant.*

*Proof.* We first show that $\xi_t$ is a subgaussian random variable.

**Claim 2.9.** *The random vector $\xi_t$ is $(16/\gamma)$-subgaussian.*

*Proof of Claim 2.9.* Note that $\xi_t = (\mathbf{g}(\mathbf{w}^t, \mathbf{w}^t, \mathbf{x}^{(t)}, y^{(t)}) - \mathbf{E}_{(\mathbf{x},y)\sim D}[\mathbf{g}(\mathbf{w}^t, \mathbf{w}^t, \mathbf{x}, y)]) \cdot (\mathbf{w}^t - \mathbf{w}^*)$ and that by construction $\|\mathbf{g}(\mathbf{w}^t, \mathbf{w}^t, \mathbf{x}, y)\|_2 \leq 4/\gamma$. Therefore, it holds that $|\mathbf{g}(\mathbf{w}^t, \mathbf{w}^t, \mathbf{x}^{(t)}, y^{(t)}) \cdot (\mathbf{w}^t - \mathbf{w}^*)| \leq 8/\gamma$, where we used that $\|\mathbf{w}^t - \mathbf{w}^*\|_2 \leq 2$ as both of these vectors lie in the unit ball. Hence, by Fact 2.7, we have that $\xi_t$ is $(16/\gamma)$-subgaussian. $\qquad\square$

Using Claim 2.9 and Definition 2.6 with parameter $\lambda = -2\lambda_t$ and $X = \xi_t$, we have that

$$\log \mathbf{E}[\exp(\widehat{V}_t)] = \log \mathbf{E}[\exp(-2\lambda_t \xi_t)] \leq C(\lambda_t^2/\gamma^2) \,,$$

where $C > 0$ is a universal constant. To bound the contribution of $\sum_{t=0}^{T} \widehat{V}_t$, we use Markov's inequality with respect to the filtration $\mathcal{F}_1, \ldots, \mathcal{F}_T$. We have that for any $Z \in \mathbb{R}$, it holds that

$$\begin{aligned}
\mathbf{Pr}_{\mathcal{Z}^1, \ldots, \mathcal{Z}^T \sim D}\left[\sum_{t=0}^{T} \widehat{V}_t \geq Z\right] &= \mathbf{Pr}_{\mathcal{Z}^1, \ldots, \mathcal{Z}^T \sim D}\left[\exp\left(\sum_{t=0}^{T} \widehat{V}_t\right) \geq \exp(Z)\right] \\
&\leq \mathbf{E}_{\mathcal{Z}^1, \ldots, \mathcal{Z}^T \sim D}\left[\exp\left(\sum_{t=0}^{T} \widehat{V}_t\right)\right] \exp(-Z) \\
&= \prod_{t=1}^{T} \mathbf{E}_{\mathcal{Z}^t \sim D}\left[\exp \widehat{V}_t \mid \mathcal{F}_t\right] \exp(-Z) \leq \exp\left(C \sum_{t=0}^{T} \frac{\lambda_t^2}{\gamma^2} - Z\right) \,,
\end{aligned}$$

where in the second inequality we use the independence of $\widehat{V}_t$ with $\{\widehat{V}_k\}_{k=1}^{t-1}$ with respect to the filtration $\mathcal{F}_t$. Recalling that $\lambda_t = c\gamma^2\epsilon$, where $c > 0$ is a sufficiently small universal constant, we have that

$$\mathbf{Pr}_{\mathcal{Z}^1, \ldots, \mathcal{Z}^T \sim D}\left[\sum_{t=0}^{T} \widehat{V}_t \geq Z\right] \leq \exp\left(Cc^2\gamma^2\epsilon^2 T - Z\right) \leq \exp\left(Cc^2\gamma^2\epsilon^2 T - Z\right) \,.$$

Setting $Z = Cc^2\gamma^2\epsilon^2 T + \log(1/\delta)$ and taking $c$ to be a sufficiently small absolute constant (as is done in our algorithm), we get that $\mathbf{Pr}_{\mathcal{Z}^1, \ldots, \mathcal{Z}^T \sim D}\left[\sum_{t=0}^{T} \widehat{V}_t \geq Z\right] \leq \delta$. This completes the proof of Lemma 2.8. $\qquad\square$

Assume that until the round $T$ the event $H_T$ holds, i.e., for all $i \in [T]$ we have that $\mathrm{err}_D(\mathbf{w}^i) \geq \eta + \epsilon$. Using Lemma 2.8 onto Inequality (7), with probability at least $1 - \delta$, we have that:

$$\begin{aligned}
\|\mathbf{w}^{T+1} - \mathbf{w}^*\|_2^2 &\leq \|\mathbf{w}^0 - \mathbf{w}^*\|_2^2 - \sum_{t=0}^{T} \lambda_t(\mathrm{err}_D(\mathbf{w}^t) - \eta) + \sum_{t=0}^{T} \widehat{V}_t \\
&\leq \|\mathbf{w}^0 - \mathbf{w}^*\|_2^2 - cT\epsilon^2\gamma^2 + \log(1/\delta) \,.
\end{aligned}$$

Running the algorithm for $T = \Theta(\log(1/\delta)/(\epsilon^2\gamma^2))$ iterations guarantees that with probability at least $1 - \delta$, we will have that $\|\mathbf{w}^{T+1} - \mathbf{w}^*\|_2^2 \leq 0$, which means $\mathbf{w}^{T+1} = \mathbf{w}^*$. In that case, i.e., in the case where all the events $H_i$ for $i \in [T]$ hold, $\mathbf{w}^{T+1}$ achieves the same error as the optimal halfspace, thus it has 0-1 error of at most $\eta + \epsilon$. Therefore, at least one vector $\mathbf{w}^{t'}$ with $t' \in [T+1]$ achieves 0-1 error of at most $\eta + \epsilon$. The algorithm, in Step (5), returns a vector $\widehat{\mathbf{w}}$ that has 0-1 error at most $\mathrm{err}_D(\widehat{\mathbf{w}}) \leq \min_{t \in [T+1]} \mathrm{err}_D(\mathbf{w}^t) + \epsilon \leq \eta + 2\epsilon$. The algorithm requires $N = O(\log(T/\delta)/(\epsilon(1 - 2\eta)))$ samples for Step (5), due to [MN06]. The algorithm draws a sample in each round and runs for at most $T$ rounds. Therefore, Algorithm 1 draws $n = N + T = \widetilde{O}(\log(1/\delta)/(\epsilon^2\gamma^2))$ samples. The algorithm needs to test each of the $T$ hypotheses with $N$ samples to find the closest one. Therefore, the total runtime is $O(dTN)$ (as in the other subroutines the algorithm uses the samples only to estimate the gradients $\mathbf{g}$, which requires $O(1)$ additions of $d$-dimenional vectors). This completes the proof of Theorem 2.1. $\qquad\square$

## 3 Conclusions and Open Problems

In this paper, we give the first sample near-optimal and computationally efficient algorithm for learning margin halfspaces in the presence of Massart noise. Specifically, the sample complexity of our algorithm nearly matches the computational sample complexity of the problem and its computational complexity is polynomial in the sample size. An interesting direction for future work is to develop a sample near-optimal and computationally efficient learner for general halfspaces (i.e., without the margin assumption). While our approach can likely be leveraged to obtain an efficient algorithm with sample complexity $\mathrm{poly}(d)/\epsilon^2$, the sample dependence on the dimension $d$ would be suboptimal. Obtaining the right dependence on the dimension seems to require novel ideas, as prior works rely on fairly sophisticated methods [DV04, DKT21, DTK23] to effectively reduce to the large margin case.

## Acknowledgments

ID was supported in part by NSF Medium Award CCF-2107079 and an H.I. Romnes Faculty Fellowship. NZ was supported in part by NSF Medium Award CCF-2107079.

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

# Supplementary Material

**Organization** The structure of this appendix is as follows: In Appendix A, we provide additional summary and comparison with related and prior work. In Appendix B, we provide a polynomial time cutting-planes based algorithm with sample complexity $\widetilde{O}(1/(\epsilon^2\gamma^4))$. Finally, in Appendix C, we provide the proofs omitted from Section 2.

# A    Related and Prior Work

## A.1    Additional Related Work

The computational problem of learning halfspaces with Massart noise has been extensively studied, both in the distribution-specific and the distribution-free settings.

In the distribution-specific setting, the first efficient algorithm for homogeneous Massart halfspaces was given in [ABHU15]. Subsequent work generalized this result in various directions [ABHZ16, ZLC17, YZ17, DKTZ20a, DKTZ20b, DKK$^+$20, DKK$^+$21, DKK$^+$22].

The first algorithmic progress in the distribution-free setting was made by [DGT19], answering a longstanding open problem [Slo88, Slo92, Blu03]. Subsequent work gave an algorithm with improved sample complexity [CKMY20] and provided strong evidence that an error of $\eta + \epsilon$ is the best to hope for in polynomial time [DK22, NT22, DKMR22] (in both the Statistical Query model and under plausible cryptographic assumptions). In a related direction, [DIK$^+$21] gave the first efficient boosting algorithm in the presence of Massart noise, which can boost a weak learner to one with error $\eta + \epsilon$. Finally, we note that natural generalizations of the Massart model to learning real-valued functions (in an essentially distribution-free setting) have also been studied [CKMY21, DPT21, DKRS22].

Very recent work [DDK$^+$23a] gave SQ (and low-degree polynomial testing) lower bounds for learning $\gamma$-margin halfspaces with RCN [AL88], which is a special case of Massart noise. Specifically, [DDK$^+$23a] showed that any efficient SQ algorithm for the problem requires sample complexity $\Omega(1/(\gamma^{1/2}\epsilon^2))$. Subsequently, [DDK$^+$23b] showed a related SQ lower bound under the Gaussian distribution, which can be adapted to obtain a lower bound of $\Omega(1/(\gamma\epsilon^2))$ for the margin setting.

## A.2    Comparison with [DKTZ24]

The work [DKTZ24] uses a similar sequence of loss functions for the problem of "online learning" Massart margin halfspaces. Intuitively, their goal is to minimize regret in an adversarial online setting. In their online setting, the adversary in each round commits to covariates $\mathbf{x}^1, \mathbf{x}^2 \in \mathbb{R}^d$ and distribution $D^t$ over $\mathbb{R}_+ \times \mathbb{R}_+$. Then the algorithm observes the covariates, chooses an action $a \in \{1, 2\}$, and observes a reward $r_a \in \mathbb{R}_+$. It is only guaranteed that there exists a unit vector $\mathbf{w}^*$ so that $\mathbf{E}_{(r_1, r_2) \sim D^t}[\mathrm{sign}(\mathbf{w}^* \cdot \mathbf{x}^1 - \mathbf{w}^* \cdot \mathbf{x}^2)(r_a - r_b)] \geq \Delta$ for some $\Delta > 0$.

Despite this superficial similarity, the work of [DKTZ24] has no new implications on the sample complexity of PAC learning Massart halfspaces with a margin. Specifically, they achieve a regret bound of $O(T^{3/4}/\gamma)$. If one translates this bound to a sample complexity upper bound for PAC learning, one would obtain a bound of $\Omega(1/(\epsilon^4\gamma^8))$ — which is quantitatively worse than prior work of [DGT19, CKMY20].

At a technical level, our work leverages this sequence of loss functions as subgradients of the potential function $\Phi(\mathbf{w}) = \|\mathbf{w} - \mathbf{w}^*\|_2^2$. Via a novel analysis, we show that these subgradients $\Omega(\epsilon)$-correlate with the direction of $\mathbf{w} - \mathbf{w}^*$. This in turn means that we can expect a decrease of order $\Omega(\lambda\epsilon)$ in each iteration, where $\lambda$ is the corresponding step-size, as long as we get 0-1 error more than $\eta + \epsilon$. This structural understanding suffices for obtaining an algorithm, based on a separation oracle, that achieves a sample complexity of $\widetilde{O}(1/(\gamma^4\epsilon^2))$. In order to obtain an algorithm with near-optimal sample complexity (and runtime), we required additional new ideas as elaborated in the body of the paper.

# B  Learning Margin Massart Halfspaces via Cutting Planes

In this section, we show how to use the cutting-planes method along with Lemma 2.2 to efficiently learning margin Massart Halfspaces using $\widetilde{O}(1/(\gamma^4\epsilon^2))$ samples.

Specifically, we establish the following result:

**Theorem B.1** (Learning Margin Massart Halfspaces with Cutting Planes). *Let $D$ be a distribution on $\mathbb{S}^{d-1} \times \{\pm 1\}$ which satisfies the $\eta$-Massart noise condition with respect to the $\gamma$-margin halfspace $f(\mathbf{x}) = \mathrm{sign}(\mathbf{w}^* \cdot \mathbf{x})$. Given $N = \Theta(\log(1/(\gamma\delta)/(\gamma^4\epsilon^2))$ i.i.d. samples from $D$, there is a $\mathrm{poly}(d,N)$ time algorithm that returns a vector $\hat{\mathbf{w}}$ such that $\mathrm{err}_D(\hat{\mathbf{w}}) \leq \eta + \epsilon$ with probability at least $1 - \delta$.*

**Remark B.2.** We can always assume that $d = \widetilde{O}(1/\gamma^2)$. This holds since we can efficiently preprocess the data, using the Johnson-Lindenstrauss transform [JL84]. Similar dimension-reduction steps have been use in prior work, e.g., [CKMY20, DDK+23a].

Given the above remark, it suffices to establish the following:

**Theorem B.3.** *Let $D$ be a distribution on $\mathbb{S}^{d-1} \times \{\pm 1\}$ which satisfies the $\eta$-Massart noise condition with respect to the $\gamma$-margin halfspace $f(\mathbf{x}) = \mathrm{sign}(\mathbf{w}^* \cdot \mathbf{x})$. Given $N = \Theta(d\log(1/(\gamma\delta)/(\gamma^2\epsilon^2))$ i.i.d. samples from $D$, there is a $\mathrm{poly}(d,N)$ time algorithm that returns a vector $\hat{\mathbf{w}}$ such that $\mathrm{err}_D(\hat{\mathbf{w}}) \leq \eta + \epsilon$ with probability at least $1 - \delta$.*

The idea of using the cutting plane method is slightly adapted from [CKMY20]. Given access to a separation oracle for a convex set $\mathcal{K}$, we can find a point inside the set $\mathcal{K}$ by querying the separation oracle $O(d\log d)$ times. The difference with [CKMY20] is that we are using a more sophisticated (and sample efficient) separation oracle. This allows us to use $O(1/\epsilon^2)$ samples, instead of $O(1/\epsilon^3)$ samples, and leads to the optimal sample complexity as a function of $\epsilon$ (but not $\gamma$).

**Fact B.4.** *Suppose that $\mathcal{K}$ is an (unknown) convex body in $\mathbb{R}^d$ which contains a Euclidean ball of radius $r > 0$ and contained in a Euclidean ball centered at the origin of radius $R > 0$. There exists an algorithm which, given access to a separation oracle for $\mathcal{K}$, finds a point $\mathbf{x}^* \in \mathcal{K}$, runs in time $\mathrm{poly}(\log(R/r), d)$, and makes $O(d\log(Rd/r))$ calls to the separation oracle.*

We first show that if we get enough samples, we can efficiently approximate the gradients $\mathbf{G}(\mathbf{w}, \mathbf{w})$. Formally, we have:

**Proposition B.5** (Separation Oracle). *Let $\epsilon, \delta \in (0, 1)$ and let $D$ be a distribution on $\mathbb{S}^{d-1} \times \{\pm 1\}$ satisfying the $\eta$-Massart noise condition with respect to the halfspace $f(\mathbf{x}) = \mathrm{sign}(\mathbf{w}^* \cdot \mathbf{x})$. Fix $\mathbf{w} \in \mathbb{R}^d$ with $\|\mathbf{w}\|_2 \leq 1$. Let $N \gtrsim \log(1/(\gamma\delta))/(\epsilon^2\gamma^2))$ and $\widehat{D}_N$ be the corresponding empirical distribution. Then, with probability at least $1 - \delta$, it holds that*

$$\mathbf{G}_{\widehat{D}_N}(\mathbf{w}, \mathbf{w}) \cdot (\mathbf{w} - \mathbf{w}^*) \geq 2(\mathrm{err}_D(\mathbf{w}) - \eta) - \epsilon \ .$$

*Proof.* By construction, $\mathbf{G}_{\widehat{D}_N}(\mathbf{w}, \mathbf{w}) = \mathbf{G}^1_{\widehat{D}_N}(\mathbf{w}) + \mathbf{G}^2_{\widehat{D}_N}(\mathbf{w})$ and by Lemma 2.2 we have that $\mathbf{G}^1_{\widehat{D}_N}(\mathbf{w}) \cdot (\mathbf{w} - \mathbf{w}^*) \geq 2(\mathrm{err}_{\widehat{D}_N}(\mathbf{w}) - \eta)$. By definition, we have $\mathbf{E}_{(\mathbf{x}^{(1)},y^{(1)}),\ldots,(\mathbf{x}^{(N)},y^{(N)})\sim D}[\mathbf{G}^2_{\widehat{D}_N}(\mathbf{w})] = 0$, where the expectation is taken with respect to the sample set. Note that the norm of $\mathbf{g}^1(\mathbf{w}, \mathbf{x}), \mathbf{g}^2(\mathbf{w}, \mathbf{x}, y)$, i.e., $\|\mathbf{g}^1(\mathbf{w}, \mathbf{x})\|_2, \|\mathbf{g}^2(\mathbf{w}, \mathbf{x}, y)\|_2$, is bounded pointwise from above by $4/\gamma$ for all $\mathbf{w} \in \mathbb{R}^d$. This can be seen as $\|\mathbf{x}\|_2 \leq 1$, $W(\cdot, \gamma/2) \leq 2/\gamma$, and $(1 - 2\eta), (1 - 2\eta(\mathbf{x})) \leq 1$.

We use the following concentration inequality to show that our sample size is enough to guarantee that the estimated gradient is close to its population version.

**Fact B.6** ([SZ07], Lemma 1). *Let $\mathbf{Z}_1, \ldots, \mathbf{Z}_n \in \mathbb{R}^d$ be random vectors such that for each $i \in [n]$ it holds $\|\mathbf{Z}_i\|_2 \leq M < \infty$ almost surely and let $\sigma^2 = \sum_{i=1}^n \mathbf{E}[\|\mathbf{Z}_i\|_2^2]$. Then, we have that for any $\epsilon > 0$,*

$$\mathbf{Pr}\left[\left\|\frac{1}{n}\sum_{i=1}^n (\mathbf{Z}_i - \mathbf{E}[\mathbf{Z}_i])\right\|_2 \geq \epsilon\right] \leq 2\exp\left(-\frac{n\epsilon}{2M}\log\left(1 + \frac{nM\epsilon}{\sigma^2}\right)\right) \ .$$

Using Fact B.6, along with the inequality $\log(1 + z) \geq z/2$, for $z \in (0, 1)$, we get that if $N \geq \Theta(\frac{\log(1/\delta)}{(\epsilon\gamma)^2})$, with probability at least $1 - \delta$, we have

$$\left\| \mathbf{G}_{\widehat{D}_N}^1(\mathbf{w}) - \underset{(\mathbf{x},y)\sim D}{\mathbf{E}}[\mathbf{g}^1(\mathbf{w},\mathbf{x})] \right\|_2 \leq \epsilon \, , \tag{8}$$

and

$$\left\| \mathbf{G}_{\widehat{D}_N}^2(\mathbf{w}) - \underset{(\mathbf{x},y)\sim D}{\mathbf{E}}[\mathbf{g}^2(\mathbf{w},\mathbf{x},y)] \right\|_2 \leq \epsilon \, . \tag{9}$$

To complete the proof, recall that by Lemma 2.2 it holds $\mathbf{G}_{\widehat{D}_N}^1(\mathbf{w}) \cdot (\mathbf{w} - \mathbf{w}^*) \geq 2(\mathrm{err}_{\widehat{D}_N}(\mathbf{w}) - \eta) - \epsilon$. Therefore, by taking the expectation over $D_{\mathbf{x}}$, we get that

$$\mathbf{G}_D^1(\mathbf{w}) \cdot (\mathbf{w} - \mathbf{w}^*) \geq 2(\mathrm{err}_D(\mathbf{w}) - \eta) \, .$$

The proof is completed by recalling that $\|\mathbf{G}_{\widehat{D}_N}^1(\mathbf{w}) - \mathbf{E}_{(\mathbf{x},y)\sim D}[\mathbf{g}^1(\mathbf{w},\mathbf{x})]\|_2 \leq \epsilon$ from Inequality (8) and that $\mathbf{E}_{(\mathbf{x},y)\sim D}[\mathbf{g}^2(\mathbf{w},\mathbf{x},y)] = 0$. $\qquad\square$

Equipped with Proposition B.5, we are ready to prove a weaker version of Theorem 2.1 using separation oracles and the cutting plane algorithm. Formally, we show that

*Proof of Theorem B.3.* Our convex set $\mathcal{K}$ is a Euclidean ball of radius $\gamma/2$ centered at $\mathbf{w}^*$. To see that, note that for any $\mathbf{v}$ such that $\|\mathbf{w}^* - \mathbf{v}\|_2 \leq \gamma/2$, we have that $|(\mathbf{w}^* - \mathbf{v}) \cdot \mathbf{x}| \leq \gamma/2$ for any $\mathbf{x}$ with $\|\mathbf{x}\|_2 = 1$. This implies that $\gamma/2 + \mathbf{w}^* \cdot \mathbf{x} \geq \mathbf{v} \cdot \mathbf{x} \geq \mathbf{w}^* \cdot \mathbf{x} - \gamma/2$. Moreover, by definition we have that $\mathbf{w}^* \cdot \mathbf{x} \geq \gamma$. Hence, if $\mathbf{w}^* \cdot \mathbf{x} \geq 0$, we have that $\mathbf{v} \cdot \mathbf{x} \geq \gamma/2$; and if $\mathbf{w}^* \cdot \mathbf{x} \leq 0$, we have that $\mathbf{v} \cdot \mathbf{x} \leq -\gamma/2$. Therefore, this ball contains all the vectors $\mathbf{w}$ with margin $\gamma/2$ and separates the points in the same way as $\mathbf{w}^*$.

Therefore, as long as we are not in the set $\mathcal{K}$ or the 0-1 error is more than $\eta + \epsilon$, we can use Proposition B.5 to construct a new separation oracle. By Fact B.4, the maximum number of calls to the separation oracle is $T = O(d\log(d/\gamma))$. By Proposition B.5, in each round we need $n = O(\log(T/\delta))/(\epsilon^2\gamma^2)$ samples from $D$ to construct a separation oracle. Therefore, the maximum number of samples is $O(nT) = O(d\log(T/\delta))/(\epsilon^2\gamma^2)$. This completes the proof. $\qquad\square$

# C Omitted Proofs from Section 2

## C.1 Proof of Claim C.1

**Claim C.1** (Claim 2.1 [DGT19]). *For any* $\mathbf{w}, \mathbf{x}$*, we have that*

$$\ell_\lambda(\mathbf{w}, \mathbf{x}, y) = \left(\mathbb{1}\{y(\mathbf{w} \cdot \mathbf{x}) \leq 0\} - \lambda\right)|\mathbf{w} \cdot \mathbf{x}| \, .$$

*Proof.* Recall that

$$\ell_\lambda(\mathbf{w}, \mathbf{x}, y) = \mathrm{LeakyReLU}_\lambda(-y(\mathbf{w}\cdot\mathbf{x})) = (1-\lambda)\mathbb{1}\{y(\mathbf{w}\cdot\mathbf{x}) \leq 0\}(-y\mathbf{w}\cdot\mathbf{x}) + \lambda\mathbb{1}\{y(\mathbf{w}\cdot\mathbf{x}) > 0\}(-y\mathbf{w}\cdot\mathbf{x}) \, .$$

Therefore, we have that

$$\ell_\lambda(\mathbf{w}, \mathbf{x}, y) = (1 - \lambda)\mathbb{1}\{y(\mathbf{w} \cdot \mathbf{x}) \leq 0\}|y\mathbf{w} \cdot \mathbf{x}| - \lambda\mathbb{1}\{y(\mathbf{w} \cdot \mathbf{x}) > 0\}|y\mathbf{w} \cdot \mathbf{x}|$$

$$= \mathbb{1}\{y(\mathbf{w} \cdot \mathbf{x}) \leq 0\}|\mathbf{w} \cdot \mathbf{x}| - \lambda|\mathbf{w} \cdot \mathbf{x}| = \left(\mathbb{1}\{y(\mathbf{w} \cdot \mathbf{x}) \leq 0\} - \lambda\right)|\mathbf{w} \cdot \mathbf{x}| \, ,$$

where we used that $y \in \{\pm 1\}$. $\qquad\square$

## C.2 Proof of Claim 2.3

We restate and prove the following claim:

**Claim 2.3.** *For any* $\mathbf{x}^{(i)} \in R_1$*, we have that* $\mathbf{g}^1(\mathbf{w}, \mathbf{x}^{(i)}) \cdot (\mathbf{w} - \mathbf{w}^*) \geq 2(\mathrm{err}(\mathbf{w}, \mathbf{x}^{(i)}) - \eta) \, .$

*Proof of Claim 2.3.* For any $\mathbf{x}^{(i)} \in R_1$, we have that

$$
\begin{aligned}
\mathbf{g}^1(\mathbf{w}, \mathbf{x}^{(i)}) \cdot \mathbf{w} &= \left( (1 - 2\eta)\mathrm{sign}(\mathbf{w} \cdot \mathbf{x}^{(i)}) - (1 - 2\eta(\mathbf{x}^{(i)}))\mathrm{sign}(\mathbf{w}^* \cdot \mathbf{x}^{(i)}) \right) \mathbf{w} \cdot \mathbf{x}^{(i)} W(\mathbf{w} \cdot \mathbf{x}^{(i)}) \\
&= \left( (1 - 2\eta)\mathrm{sign}(\mathbf{w} \cdot \mathbf{x}^{(i)}) - (1 - 2\eta(\mathbf{x}^{(i)}))\mathrm{sign}(\mathbf{w}^* \cdot \mathbf{x}^{(i)}) \right) \mathrm{sign}(\mathbf{w} \cdot \mathbf{x}^{(i)}) \\
&= 2(\mathrm{err}(\mathbf{w}, \mathbf{x}^{(i)}) - \eta) ,
\end{aligned} \tag{10}
$$

where we used that for any $\mathbf{x}^{(i)} \in R_1$, $W(\mathbf{w} \cdot \mathbf{x}^{(i)}) = 1/|\mathbf{w} \cdot \mathbf{x}^{(i)}|$, and hence $W(\mathbf{w} \cdot \mathbf{x}^{(i)}, \gamma/2)\mathbf{w} \cdot \mathbf{x}^{(i)} = \mathrm{sign}(\mathbf{w} \cdot \mathbf{x}^{(i)})$; and that $\mathrm{err}(\mathbf{w}, \mathbf{x}^{(i)}) = \eta(\mathbf{x}^{(i)})$ if $\mathrm{sign}(\mathbf{w} \cdot \mathbf{x}^{(i)}) = \mathrm{sign}(\mathbf{w}^* \cdot \mathbf{x}^{(i)})$ and $1 - \eta(\mathbf{x}^{(i)})$ otherwise.

We now bound the contribution of $\mathbf{w}^*$. Since $\eta(\mathbf{x}) \leq \eta$, we have

$$
(1 - 2\eta(\mathbf{x})) - (1 - 2\eta)\mathrm{sign}(\mathbf{w} \cdot \mathbf{x})\mathrm{sign}(\mathbf{w}^* \cdot \mathbf{x}) \geq 0 .
$$

Therefore, we have that

$$
\begin{aligned}
\mathbf{g}^1(\mathbf{w}, \mathbf{x}^{(i)}) \cdot \mathbf{w}^* &= \left( (1 - 2\eta)\mathrm{sign}(\mathbf{w} \cdot \mathbf{x}) - (1 - 2\eta(\mathbf{x}))\mathrm{sign}(\mathbf{w}^* \cdot \mathbf{x}) \right) \mathrm{sign}(\mathbf{w}^* \cdot \mathbf{x})|\mathbf{w}^* \cdot \mathbf{x}|W(\mathbf{w} \cdot \mathbf{x}^{(i)}) \\
&= -\left( (1 - 2\eta(\mathbf{x})) - (1 - 2\eta)\mathrm{sign}(\mathbf{w} \cdot \mathbf{x})\mathrm{sign}(\mathbf{w}^* \cdot \mathbf{x}) \right)|\mathbf{w}^* \cdot \mathbf{x}|W(\mathbf{w} \cdot \mathbf{x}^{(i)}) \leq 0 ,
\end{aligned}
$$

which gives that $-\mathbf{g}^1(\mathbf{w}, \mathbf{x}^{(i)}) \cdot \mathbf{w}^* \geq 0$. This completes the proof of Claim 2.3. $\square$

## C.3 Proof of Claim 2.4

We restate and prove the following:

**Claim 2.4.** *For any $\mathbf{x}^{(i)} \in R_2$, we have that $\mathbf{g}^1(\mathbf{w}, \mathbf{x}^{(i)}) \cdot (\mathbf{w} - \mathbf{w}^*) \geq 2(\mathrm{err}(\mathbf{w}, \mathbf{x}^{(i)}) - \eta) .$*

*Proof of Claim 2.4.* We have that

$$
\begin{aligned}
\mathbf{g}^1(\mathbf{w}, \mathbf{x}^{(i)}) \cdot (\mathbf{w} - \mathbf{w}^*) &= \left( (1 - 2\eta)\mathrm{sign}(\mathbf{w} \cdot \mathbf{x}^{(i)}) - (1 - 2\eta(\mathbf{x}^{(i)}))\mathrm{sign}(\mathbf{w}^* \cdot \mathbf{x}^{(i)}) \right) \left( \frac{\mathbf{w} \cdot \mathbf{x}^{(i)} - \mathbf{w}^* \cdot \mathbf{x}^{(i)}}{\max(\gamma/2, |\mathbf{w} \cdot \mathbf{x}^{(i)}|)} \right) \\
&= \left( (1 - 2\eta)\mathrm{sign}(\mathbf{w} \cdot \mathbf{x}^{(i)}) - (1 - 2\eta(\mathbf{x}^{(i)}))\mathrm{sign}(\mathbf{w}^* \cdot \mathbf{x}^{(i)}) \right) \left( \frac{\mathbf{w} \cdot \mathbf{x}^{(i)} - \mathbf{w}^* \cdot \mathbf{x}^{(i)}}{\gamma/2} \right) ,
\end{aligned}
$$

where we used that $\max(\gamma/2, |\mathbf{w} \cdot \mathbf{x}^{(i)}|) = \gamma/2$ for any $\mathbf{x}^{(i)} \in R_2$. Since $\mathrm{sign}(\mathbf{w}^* \cdot \mathbf{x})$ has $\gamma$-margin, we have that $|\mathbf{w}^* \cdot \mathbf{x}^{(i)}| \geq \gamma$. Since $\mathbf{x}^{(i)} \in R_2$, it holds $|\mathbf{w} \cdot \mathbf{x}^{(i)}| < \gamma/2$. Therefore, $-\mathrm{sign}(\mathbf{w}^* \cdot \mathbf{x}^{(i)})(\mathbf{w} \cdot \mathbf{x}^{(i)} - \mathbf{w}^* \cdot \mathbf{x}^{(i)}) = (|\mathbf{w}^* \cdot \mathbf{x}^{(i)}| - \mathrm{sign}(\mathbf{w}^* \cdot \mathbf{x}^{(i)})\mathbf{w} \cdot \mathbf{x}^{(i)}) \geq \gamma/2$. This in turn implies that

$$
\begin{aligned}
\mathbf{g}^1(\mathbf{w}, \mathbf{x}^{(i)}) \cdot (\mathbf{w} - \mathbf{w}^*) &\geq (1 - 2\eta(\mathbf{x}^{(i)}) - (1 - 2\eta)\mathrm{sign}(\mathbf{w} \cdot \mathbf{x}^{(i)})\mathrm{sign}(\mathbf{w}^* \cdot \mathbf{x}^{(i)})) \\
&= 2(\mathrm{err}(\mathbf{w}, \mathbf{x}^{(i)}) - \eta) ,
\end{aligned}
$$

completing the proof of Claim 2.4. $\square$

