# OpenReview forum: "A Near-optimal Algorithm for Learning Margin Halfspaces with Massart Noise"
_NeurIPS.cc/2024/Conference — NeurIPS 2024 spotlight_

### Official Review · Reviewer_2xNf · 2024-06-27

**Soundness:** 3
**Presentation:** 3
**Contribution:** 3
**Rating:** 7
**Confidence:** 3

**Summary:**

The paper considers the problem of PAC learning halfspaces with margin in the presence of Massart noise. The paper provides an algorithm that well-balances sample and computational efficiency. Specifically, the dependence of the algorithm on both $\epsilon$ and $\gamma$ is near-optimal.

**Strengths:**

1. The studied problem is fundamental in the area of PAC learning, and the paper provides a significant progress on it.
2. The paper is well written, and the main techniques are well explained.

**Weaknesses:**

1. Results might be somewhat weaker than presented (see questions), and some phrasings in this context are too vague (for example "there is evidence that...").
2. The natural agnostic extension of the problem is not discussed.

**Questions:**

Questions:
1. I'm not sure about the statement "...we essentially settle this question..."  in line 71. As far as I understand, the optimal computational sample complexity might be $\tilde{O}(1/\epsilon^2 + 1/\epsilon \gamma^2)$ and not $\tilde{\Omega}(1/\epsilon^2 \gamma^2)$. Either way, the provided upper bound is impressive enough to recommend for acceptance.
2. Is there a specific reason not to discuss the agnostic case? Is it usualy considered in Massart noise problems?


Suggestions:

Consider writing what is $\eta$ in the abstract.

**Limitations:**

Yes.

---

> ### Author Rebuttal · Authors · 2024-08-07
>
> We thank the reviewer for the time and effort in reading our paper and the positive assessment.  We respond to each point raised by the reviewer below.
>
> >(Weakness 1): Results might be somewhat weaker than presented (see questions), and some phrasings in this context are too vague (for example "there is evidence that...").
>
> *Response:*  We thank the reviewer for this feedback. We will make our statements of prior information-computation tradeoffs precise. In the submitted version, we did not discuss these results in detail due to space limitations. Please see response below for more details.
>
> >(Weakness 2): The natural agnostic extension of the problem is not discussed.
>
> *Response:*   For distribution-free agnostic PAC learning, the learning problem we study is known to be computationally intractable (even for weak learning). Specifically, it is is NP-Hard [1] for proper learning and cryptographically/SQ-hard for improper learning [2,3]. These hardness results have historically been one of the motivations for studying the Massart model.
>
> >(Question 1): I'm not sure about the statement "...we essentially settle this question..."  in line 71. As far as I understand, the optimal computational sample complexity might be $\tilde{O}(1/\epsilon^2 + 1/\epsilon \gamma^2)$ and not $\tilde{\Omega}(1/\epsilon^2 \gamma^2)$. Either way, the provided upper bound is impressive enough to recommend for acceptance.
>
> *Response:* As the reviewer points out, the best known lower bound on the computational sample complexity of the problem is  $\tilde{\Omega}(1/(\gamma\epsilon^2) + 1/(\epsilon \gamma^2))$. This lower bound is quadratic in both parameters of interest but does not quite match our upper bound. While we believe that a lower bound of order $\tilde{\Omega}(1/(\gamma^2\epsilon^2))$ exists, this remains an open problem. Finally, we note that even for the easier model of Random Classification Noise (RCN), the best known efficient algorithm has sample complexity $\tilde{O}(1/(\gamma^2\epsilon^2))$ (established recently in [DDK+23a]).
>
> >(Question 2): Is there a specific reason not to discuss the agnostic case? Is it usually considered in Massart noise problems?
>
>
> *Response:* See Response to  Weakness 2 above.
>
>
>
> > Suggestions: Consider writing what is $\eta$ in the abstract.
>
> *Response:* Thank you. We will add this in the final version.
>
> [1] Hardness of Learning Halfspaces with Noise, Venkatesan Guruswami, Prasad Raghavendra, FOCS 2006
>
> [2] Complexity Theoretic Limitations on Learning Halfspaces, Amit Daniely, STOC 2016
>
> [3] Hardness of agnostically learning halfspaces from worst-case lattice problems, Stefan Tiegel, COLT 2023

---

> > ### Comment · Reviewer_2xNf · 2024-08-08
> >
> > Thank you for addressing my comments and questions.

---

### Official Review · Reviewer_Sz9S · 2024-07-05

**Soundness:** 3
**Presentation:** 3
**Contribution:** 3
**Rating:** 7
**Confidence:** 4

**Summary:**

The paper considers the problem of PAC-learning $\gamma$-margin halfspaces under $\eta$-Massart noise. The paper provides an efficient algorithm achieving error $\eta+\epsilon$ with sample complexity $\tilde{O}(1/(\epsilon^2\gamma^2))$. Since previous work provided evidence that an inverse-quadratic dependence on $\epsilon$ is necessary for efficient algorithms, the algorithm appears to be near-optimal in terms of sample-complexity up to logarithmic factors.

The algorithm relies on an iterative stochastic-gradient descent approach, where at each iteration a new loss function is defined which determines the gradient descent of the next iteration. The paper proves that if $T$ iterations are performed (where $T$ is a sufficiently large multiple of $\log(1/\delta)/(\epsilon^2\gamma^2)$), then with probability at least $1-\delta$, one of the halfspaces obtained in $T$ iterations have error at most $\eta+\epsilon$. By drawing some extra independent samples and computing the empirical error for each of these halfspaces, one can find a good halfspace.

The paper is generally well written.

**Strengths:**

The paper provides a near-optimal algorithm (in terms of sample complexity) for learning $\gamma$-margin halfspaces under Massart noise.

**Weaknesses:**

I haven't found significant weaknesses in the paper.

Typos:
- Page 2, line 34: “We say that that the distribution” ->  “We say that the distribution”
- Page 5: Line 195: If I am not mistaken, the function g is twice the gradient.

**Questions:**

- The authors claim that the computational complexity of the algorithm is linear in the samples. However, it seems to me that the complexity of step (5) is $O(T\cdot N\cdot d)$. It does not seem to be trivial to implement step (5) in $O((T + N)\cdot d)$ time.

- While previous work show that $\Omega(1/(\gamma^2\epsilon))$ and $\Omega(1/\epsilon^2)$ are lower bounds, it doesn't necessarily follow that  $\Omega(1/(\epsilon^2\gamma^2))$ is a lower bound as well.

**Limitations:**

Since this is a theoretical paper, I can't see any potential negative societal impact.

---

> ### Author Rebuttal · Authors · 2024-08-07
>
> We thank the reviewer for the time and effort in reading our paper and the positive assessment. We respond to each point raised by the reviewer below.
>
> >(Question 1): The authors claim that the computational complexity of the algorithm is linear in the samples. However, it seems to me that the complexity of step (5) is $O(T\cdot N\cdot d)$. It does not seem to be trivial to implement step (5) in $O((T + N)\cdot d)$ time.
>
>
> *Response:* Yes, the reviewer is correct. We will fix the statement about the runtime where appropriate. As the reviewer notes, the running time incurs an extra $1/\epsilon$ multiplicative term (up to logarithmic factors). That is because $N$ is set to $\tilde{O}(\log(1/(\delta\gamma))/(\epsilon (1-2\eta)))$ and we need to evaluate each of the $T$ hypotheses to return the best one.
>
> >(Question 2):  While previous work show that $\Omega(1/(\gamma^2\epsilon))$ and $\Omega(1/\epsilon^2)$ are lower bounds, it doesn't necessarily follow that $\Omega(1/(\epsilon^2\gamma^2))$ is a lower bound as well.
>
>
> *Response:* We agree with the reviewer about prior work on hardness, and we will adjust our phrasing. In more detail, the best known lower bound for the computational sample complexity of this problem is $\tilde{\Omega}( 1/(\epsilon \gamma^2)+1/(\gamma\epsilon^2) )$, and applies to SQ algorithms and low-degree polynomial tests. The first term is the information-theoretic sample complexity [MN06] and the second term follows from [DDK+23a,DDK+23b].
>
> While we have good reason to believe that these hardness results can be improved to give a lower bound of $\Omega(1/(\epsilon^2\gamma^2))$, this remains an open problem.

---

> > ### Comment · Reviewer_Sz9S · 2024-08-11
> >
> > I would like to thank the authors for their response. Since they addressed my minor comments, I'm increasing my score to 7.

---

### Official Review · Reviewer_JiXV · 2024-07-12

**Soundness:** 4
**Presentation:** 4
**Contribution:** 3
**Rating:** 7
**Confidence:** 3

**Summary:**

This paper studies the problem of PAC learning $\gamma$-margin halfspaces under Massart noise: to PAC learn any distribution $D$ such that there exists $w^*$ with the bounded norm of $1$ that has a margin of at least $\gamma$, i.e. $\mathbb{P}_{(x,y)\sim D}( |\langle w^*,x \rangle| \geq \gamma)=1$ and for every $x$, we have $\mathbb{P}(\text{sign}(\langle w^*,x \rangle) \neq y )=\eta(x)$, where a function $\eta$ is bounded in $[0,\eta]$ (with overload of notation).

This is a classical problem in learning theory. Information theoretically, one needs $\tilde{\Theta}(1/\gamma^2 \epsilon)$ samples, while for computationally efficient algorithms, it is widely believed that inverse quadratic dependence in $\epsilon$ (e.g. $1/\epsilon^2$) is necessary.

The main result of this paper is to present an efficient algorithm with nearly optimal sample complexity $\tilde{O}(1/\gamma^2 \epsilon^2)$, essentially closing the question.

**Strengths:**

1. The paper studies a problem that is important to the learning theory community and provides a state-of-the-art result. The previous best algorithm required $O(1/\gamma^4 \epsilon^3)$ samples.
2. The paper is well-written.

**Weaknesses:**

I do not see any major weaknesses.

**Questions:**

None.

**Limitations:**

This is a theoretical work with no societal impact.

---

> ### Author Rebuttal · Authors · 2024-08-07
>
> We thank the reviewer for the time and effort in reviewing our paper and the positive feedback.

---

### Official Review · Reviewer_SyXT · 2024-07-13

**Soundness:** 4
**Presentation:** 4
**Contribution:** 3
**Rating:** 7
**Confidence:** 3

**Summary:**

The paper studies the problem of learning a $\\gamma$-margin halfspace with $\\eta$ Massart noise and provides the first computationally efficient algorithm having 0-1 error $<= \\eta + \\varepsilon$ with sample complexity $O(1/\\gamma^2\\varepsilon^2)$ which nearly matches the information theoretic bound of $O(1/\\gamma^2\\varepsilon^)$ improving upon the previous $O(1/\\gamma^4\\varepsilon^3)$ sample complexity by [Chen et al. ‘20].
The main contributions of the paper are a novel “global” (as opposed to previous conditioning based) optimization to solve this problem directly using gradient descent. Towards this, a novel convex loss is defined using an independent vector as a parameter, and using the margin as a bound on the scaling parameter. The gradient w.r.t. the solution vector is used in an SGD loop, while the independent vector is updated by the gradient step, and its projection on the unit ball is the updated solution. This clever formulation leads to the gradient being composed of the “correct” direction of optimization along with an estimation error. The former minimizes the distance to the true solution, while the latter is bounded using standard concentration arguments.

**Strengths:**

1. Novel loss formulation and gradient update step which can be directly used via SGD.
2. Simple algorithm and analysis.
3. Near optimal computational bound on an important problem.

**Weaknesses:**

Result is specific to a particular noise model and it is not clear if the techniques are more broadly applicable.

**Questions:**

In Algorithm 1, $\\lambda_t$ seems to be a constant independent of $t$, so it can be fixed outside the loop.

**Limitations:**

Yes

---

> ### Author Rebuttal · Authors · 2024-08-07
>
> We thank the reviewer for the positive feedback and the provided questions. We respond to each point raised by the reviewer below.
>
> > (Weaknesses 1):Result is specific to a particular noise model and it is not clear if the techniques are more broadly applicable.
>
> *Response:* We would like to point out that the Massart (or bounded noise) model is essentially the strongest label noise model that allows for polynomial-time algorithms in the distribution-free PAC setting. In particular, if the label noise is fully adversarial (agnostic model), it is computationally hard to achieve any non-trivial error guarantees for the class of margin halfspaces (aka to achieve "weak learning"); see [1,2,3]. Moreover, we believe that the problem we study and the results themselves are interesting in their own right (the algorithmic complexity of the problem has been a longstanding open question; the NeurIPS'19 paper on the topic received the best paper award at that conference and has subsequently had a significant impact). That said, while our focus has been on this particular problem, we believe that the technical analysis of our algorithm could be of broader interest. Specifically, we feel that our white-box analysis of online SGD is novel and could be useful elsewhere. Moreover, the reweighting scheme that we employ may be useful in other problems, as it provides a method to *convexify* the 0-1 loss.
>
> > (Question): In Algorithm 1, $\lambda_t$ seems to be a constant independent of $t$, so it can be fixed outside the loop.
>
> *Response:*  Thank you for pointing this out. We will move $\lambda_t$ outside the main loop.
>
>
>
> [1] Hardness of Learning Halfspaces with Noise, Venkatesan Guruswami, Prasad Raghavendra, FOCS 2006
>
> [2] Complexity Theoretic Limitations on Learning Halfspaces, Amit Daniely, STOC 2016
>
> [3] Hardness of agnostically learning halfspaces from worst-case lattice problems, Stefan Tiegel, COLT 2023

---

> > ### Comment · Reviewer_SyXT · 2024-08-11
> >
> > Thank you  for the rebuttal and the clarification. I will keep my rating.

---

### Official Review · Reviewer_g5LR · 2024-07-15

**Soundness:** 4
**Presentation:** 4
**Contribution:** 3
**Rating:** 8
**Confidence:** 3

**Summary:**

This paper essentially resolves the *computational sample complexity* of learning $\gamma$-margin halfspaces under the Massart noise model. Here, computational sample complexity refers to the number of samples required for polynomial-time algorithms, as opposed to general statistical estimators which may be computationally intractable. Previous works have given rigorous evidence, in the form of lower bounds against large classes of algorithms such as SQ algorithms, of an *information-computation* gap for learning $\gamma$-margin halfspaces. Without restrictions on computation, the sample complexity is $\Theta(1/(\gamma^2 \epsilon))$ for achieving zero-one loss of $\eta + \epsilon$, where $\eta$ is the uniform bound on the Massart noise. For a large class of efficient algorithms, however, the required sample complexity is at least $\Omega(1/(\gamma^2 \epsilon^2))$.

They show that the $1/(\gamma^2 \epsilon^2)$ sample complexity is essentially tight by designing a polynomial-time algorithm for learning $\gamma$-margin halfspaces. Their algorithm is based on a novel and technically insightful choice of the loss sequence for online SGD, which results in a surprisingly simple and efficient algorithm with near-optimal sample complexity.

**Strengths:**

This is a well-written paper that stands out for being both technically insightful and simple. A simple and efficient algorithm (online SGD with a specific choice of loss sequence) achieves optimal sample complexity and lends itself to a clean analysis. It's hard to ask for more than this.

The core idea behind the algorithm is a clever choice of the loss sequence $(\ell_t)$ with respect to which one runs online SGD. Previous works have already employed the LeakyReLU loss as a convex surrogate for the zero-one loss and achieved suboptimal upper bounds on the sample complexity. LeakyReLU with parameter $\lambda > 0$ is defined by $\ell_\lambda(a) = (1-\lambda) \mathbb{1}[a \ge 0] + \lambda \mathbb{1}[a < 0]$. Applying this to the halfspace learning setting, a straightforward calculation shows that $\ell_\lambda(-y (w \cdot x)) = (\mathbb{1}[\mathrm{sign}(w\cdot x) \neq y] - \lambda)|w \cdot x|$, where $(x, y) \in \mathbb{S}^{d-1} \times \\{\pm 1\\}$ is a sample from the $\gamma$-margin halfspace distribution and $w \in \mathbb{R}^d$ is a candidate halfspace. Note the resemblance to the *shifted* zero-one loss $L(w) = \mathbb{E}_{(x,y)} \mathbb{1}[\mathrm{sign}(w\cdot x) \neq y] - \eta$ (when $\lambda = \eta$ in LeakyReLU). By the $\eta$-Massart noise assumption, $L(w^{\*}) \le 0$ for the optimal halfspace $w^{\*} \in \mathbb{R}^d$ and $L(w) \ge \epsilon$ for any halfspace $w$ with zero-one loss at least $\eta + \epsilon$.

The key difference between this shifted zero-one loss and the LeakyReLU is the $|w \cdot x|$ term. In particular, if we reweight each sample LeakyReLU loss by $1/|w \cdot x|$, we recover the shifted zero-one loss which is unfortunately non-convex. The authors overcome this issue by considering a *family* of bounded and convex loss functions $(\ell_u)$ indexed by $u \in \mathbb{R}^{d}$. Each $\ell_u$ is simply the LeakyReLU loss reweighted by $1/|u \cdot x|$. It is precisely this decoupling of the halfspace parameter $w$ and the reweighting parameter $u$ that leads to the guarantees of the algorithm. The *sequence* of reweighting parameters $(u_t)$, each of which leads to a different loss $\ell_t$, is chosen adaptively by online SGD.

**Weaknesses:**

I did not find any significant weaknesses, only minor comments regarding the presentation.

- **The expression "(vector) v is independent of w" (line 174, 176) is confusing.** I think it's easy to confuse "independence" with statistical independence. It would be helpful to clarify this by adding that the reweighting term is *constant* with respect to the parameter $w \in \mathbb{R}^d$, and the reweighting term $W(v \cdot x, \gamma)$ remains the same when taking the gradient of $L_{v}(w)$. This is already implicit in the mathematical expressions, but providing additional explanation would benefit readers.

**Questions:**

- Does the near-optimal online SGD algorithm fall within the class of SQ algorithms? If not, could there still exist non-SQ algorithms that achieve sample complexity with subquadratic dependence on $1/\epsilon$?

---

> ### Author Rebuttal · Authors · 2024-08-07
>
> We thank the reviewer for the time and effort in reviewing our paper and for the positive feedback. Below we provide specific responses to the points and questions raised by the reviewer.
>
> > (Weaknesses 1): The expression "(vector) v is independent of w" (line 174, 176) is confusing. I think it's easy to confuse "independence" with statistical independence. It would be helpful to clarify this by adding that the reweighting term is constant with respect to the parameter $w \in \mathbb{R}^d$, and the reweighting term $W(v \cdot x, \gamma)$ remains the same when taking the gradient of $L_{v}(w)$. This is already implicit in the mathematical expressions, but providing additional explanation would benefit readers.
>
> *Response:*  We thank the reviewer for this suggestion. We will clarify this point in the final version of the paper.
>
> > (Question 1) Does the near-optimal online SGD algorithm fall within the class of SQ algorithms? If not, could there still exist non-SQ algorithms that achieve sample complexity with subquadratic dependence on $1/\epsilon$?
>
> *Response:* The online GD algorithm (using a batch size) can indeed be efficiently implemented as an SQ algorithm. As we explain below, our algorithm can be formulated in this way. Therefore, the previously known SQ lower bound covers the algorithm developed in our paper. In more detail, this can be seen as follows: in each iteration of our algorithm, the update rule calculates the gradient -- which is of the form $g=\mathbf{E}[f(x,y)]$ --  where $f$ can be viewed as the query function in the SQ model.
>
> We also note that prior work established the same information-computation tradeoff for the class of low-degree polynomial tests (in addition to SQ algorithms). Historically, lower bounds against SQ algorithms and low-degree polynomials have been viewed as strong evidence of hardness. That said, these are restricted models of computation and it is in principle possible that an algorithm can surpass them.

---

> > ### Comment · Reviewer_g5LR · 2024-08-09
> >
> > Thank you for your response!

---

### Author Rebuttal · Authors · 2024-08-07

We thank the reviewers for their time, effort, and feedback. We are  encouraged by the positive comments of reviewers, and that our paper was appreciated for the: (i) **significant contribution** (SyXT, JiXV, 2xNf); (ii) **technically novel** (g5LR,SyXT) and (iii) **writing quality** (g5LR, JiXV,  2xNf). We respond to specific comments below.

---

### Decision · Program_Chairs · 2024-09-25

**Decision:**

Accept (spotlight)

**Comment:**

This paper advances state of the art results on efficient learning large-margin halfspaces with eta-Massart noise, with target error eta+epsilon; specifically its upper bound's quadratic dependences on 1/gamma and 1/epsilon are essentially not improvable in view of existing lower bounds. The reviewers also deemed the paper's technique of using online learning and convexifying the losses interesting.

The authors are encouraged to incorporate the discussion with the reviewers to clarify the running time analysis and the sense of optimality of the sample complexity bounds in the final version.